# Short-term solar energy forecasting: Integrated computational intelligence of LSTMs and GRU

Aneela Zameer[1]*, Fatima Jaffar[1], Farah Shahid[2], Muhammad Muneeb[3], Rizwan Khan[2], Rubina Nasir[4]

**1** Department of Computer & Information Sciences, Pakistan Institute of Engineering and Applied Sciences, Islamabad, Pakistan, **2** Department of Computer Science and Technology, Zhejiang Normal University, Jinhua, China, **3** Department of Mathematics, Khalifa University of Science and Technology, Abu Dhabi, United Arab Emirates, **4** Department of Physics, AIR University, Islamabad, Pakistan

\* aneelaz@pieas.edu.pk, aneelas@gmail.com

**Data Availability Statement:** "Yes - all data are fully available without restriction" "All relevant data are available from figshare at https://figshare.com/articles/dataset/_/22768967."

## Abstract

Problems with erroneous forecasts of electricity production from solar farms create serious operational, technological, and financial challenges to both Solar farm owners and electricity companies. Accurate prediction results are necessary for efficient spinning reserve planning as well as regulating inertia and power supply during contingency events. In this work, the impact of several climatic conditions on solar electricity generation in Amherst. Furthermore, three machine learning models using Lasso Regression, ridge Regression, ElasticNet regression, and Support Vector Regression, as well as deep learning models for time series analysis include long short-term memory, bidirectional LSTM, and gated recurrent unit along with their variants for estimating solar energy generation for every five-minute interval on Amherst weather power station. These models were evaluated using mean absolute error root means square error, mean square error, and mean absolute percentage error. It was observed that horizontal solar irradiance and water saturation deficiency had a highly proportional relationship with Solar PV electricity generation. All proposed machine learning models turned out to perform well in predicting electricity generation from the analyzed solar farm. Bi-LSTM has performed the best among all models with 0.0135, 0.0315, 0.0012, and 0.1205 values of MAE, RMSE, MSE, and MAPE, respectively. Comparison with the existing methods endorses the use of our proposed RNN variants for higher efficiency, accuracy, and robustness. Multistep-ahead solar energy prediction is also carried out by exploiting hybrids of LSTM, Bi-LSTM, and GRU.

## 1. Introduction

The world is consuming more energy, which raises the risk of a worldwide energy crisis with detrimental impacts on the environment [1]. Since the world's raw material supplies (fossil fuels) have significantly decreased, causing severe economic, political, and social concerns, the effective use of energy is a topic that has attracted a lot of attention [2]. It is crucial to have

**Funding:** The author(s) received no specific funding for this work.

**Competing interests:** The authors have declared that no competing interests exist.

**Abbreviations:** RMSE, root mean square error; ARIMA, Autoregressive integrated moving average; GRU, Gated recurrent unit; NN, neural network; RNN, Recurrent neural network; MAE, Mean square error; LASSO, Least Absolute Shrinkage and Selection operator; CEC, Constant error carousel; DL, Deep learning; SVR, support vector regression; LSTM, Long short-term memory; $x_t$, input; RBF, Radial basis function; $h_{t-1}$, previous LSTM block; SVM, support vector machines; DBNs, Deep Belief Networks; $u_t$, Update gate; $r_t$, Reset gate.

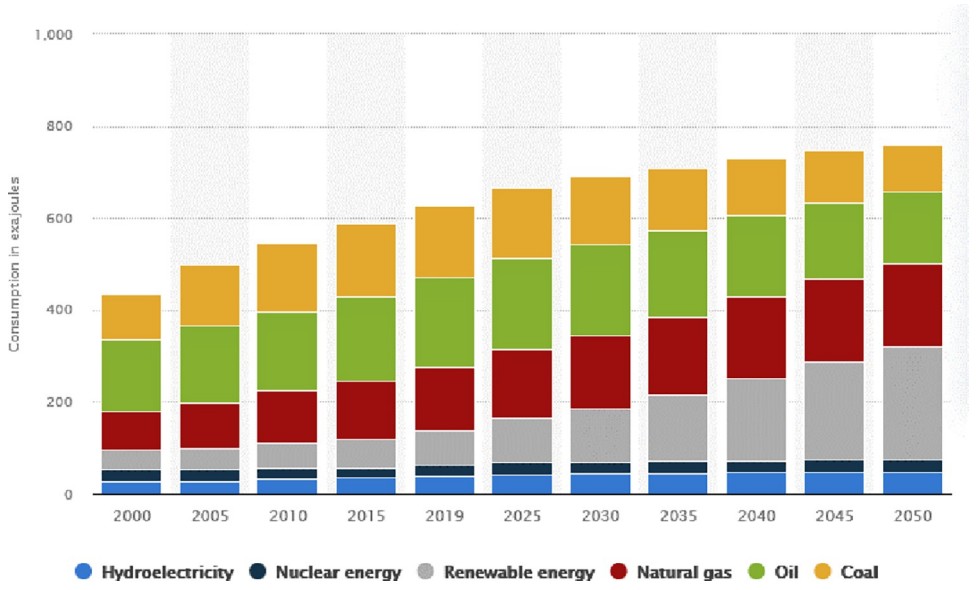

**Fig 1. World energy demand.**

accurate and persistent forecasting to improve power production from renewable energy sources such as water, wind, and solar in accordance with the electricity requirement. However, solar energy is becoming more popular as a renewable energy source than ever before, since it is one of the most environmentally friendly, cost-effective, and inexpensive choices available. Fig 1 depicts a historical forecast of worldwide demand for various types of energy, with the usage of fossil fuels continuing to lead [3]. Furthermore, the major issue of dealing with weather changes impacted the inconsistency of solar system instability. Climate change has reduced by more than 20% the quantity of power generated by photovoltaic energy plants in real-world solar electricity production [4]. As a result, these technologies are typically unable to integrate into electrical grids. To address this issue, accurate short-term forecasting of PV modules is critical for daily or hourly efficient grid production, distribution, and storage, as well as decision-making in the energy market and grid efficiency in general. The solar energy and environmental parameters in the specific location, such as temperature, humidity levels, and wind direction, influence the design of solar power-producing devices [5]. The power generated through a PV system is a stochastic variable that fluctuates over time because of the inherent fluctuation of these external factors. The difficulties in forecasting PV power output have a serious effect on areas of the power grid such as dependability, stability, planning and scheduling tasks, and market structures [6].

Therefore, one of the key research interests in the PV systems are predicting energy production. Forecasts of solar power are mostly dependent on the analysis of historical statistical data and long-term meteorological data [7], which gives vital information for forecasting expected behavior in producing systems using various approaches. Several research studies concentrate on forecasting solar irradiance employing image-based approaches, statistical features, and meteorological data simulations [8]. To compute PV power generation, the predicted solar radiation and other factors are required as input data for PV commercial modeling software programs [9].

The availability of numerical simulations for practical and accurate PV power forecasting can reduce the effects of PV uncertainty in the electricity grid while also increasing PV system implementation. Despite the enormous amount of literature on the subject, few studies

indicate the use of machine learning approaches to anticipate solar power generation [10]. Moreover, many forecasting models are built on historical data, including methods assisted by Machine Learning and Artificial Intelligence. Consequently, one of the primary research topics in PV systems is forecasting energy production. Very short-term energy prediction can improve PV module operational routing and support their assurance and crisis responses [11]. The ability of most traditional solar energy forecasting algorithms to find correlations between short amounts of data is limited; nevertheless, they are incapable of examining the correlation and uncovering implicit and relevant information about the solar energy system.

As industrialization and the contemporary way of life advance, fossil fuel use rises steadily. Fossil fuels are employed in both home and industrial settings, which has an impact on the environment's ecology and the cost of pollution. The exploitation of conventional fuels contributes both directly and indirectly to the solar system's gloomy future, along with many other issues.

Researchers and several groups are working on alternative fuels that must be abundant in nature, economically feasible, simple to use, and less polluting to remove the dependence on conventional fuels. Therefore, renewable energy sources such as solar, tidal, wind, and biofuel are preferable to traditional energy sources. Because they are environmentally benign and do not contribute to global warming, the generation of greenhouse gases, or any other negative effects, these non-conventional forms are not only renewable but also sustain ecology and the environment.

This work compares the performance of supervised learning methods for predicting solar power at Davis Meteorological Station in Amherst. LASSO, ridge regression (RR), ElasticNet regression, and support vector regressors (SVR) are used to build four different forecasting models. As well as, time series regression models of recurrent neural networks (LSTM, GRU, and Bi-LSTM) are employed to further evaluate the execution of the model. The performance of these methodologies is tested using data from the same solar energy system. The work is structured as follows. First, there is a summary of solar forecasting methods found in the literature. The solar power system and data collection process are described in the next section. The suggested methods for solar power prediction are then provided, along with a description of the performance measures. Following the discussion of the technique, the proposed hybrid algorithms LSTM_GRU, LSTM_Bi-LTM, Bi-LSTM_LSTM, Bi-LSTM_GRU, GRU_LSTM, and GRU_ Bi-LSTM examined using the collected data. Finally, several findings are presented following the corresponding discussion and analysis of the results. Salient features of our presented work are the following:

- Multivariate analysis of very-short term solar energy predictions is carried out.

- Hybrid models of time series regression were incorporated into the prediction outcomes.

- A region-wise dataset of Davis Meteorological station in Amherst is employed for the forecast.

- Multi-timestep ahead forecasting is carried out for all LSTMs and GRU techniques.

- The performance of the suggested time series models was assessed using accuracy and computational efficiency.

- Comparison with reported models and ablation strategies supports the design of individual modules in GRU.

## 2. Literature review

Solar power prediction has become a major issue in the process of increasing the penetration of solar energy sources in electric power networks due to its inconsistent nature and periodic

behavior. Accurate short-term solar power forecasting not only helps in optimizing solar energy integration into electric power grids but also assures that solar energy trades at a reasonable cost in electricity markets. Solar energy forecast, on the other hand, is highly dependent on meteorological and climatic variables in a specific area, making it a challenging problem to solve [12]. Solar photovoltaic (PV) farms have been widely erected on a vast scale all over the world, generating electricity. Germany, China, and the United States have all built large-scale photovoltaic farms, and South Korea installed around 467 megawatts (MW) of PV capacity in 2013. Germany, China, and the United States have all built large-scale photovoltaic farms. In this regard, an increase in the number of large-scale photovoltaic farms, induced in increasing the amount of solar energy in the entire electricity grid. However, the power production of these PV farms may fluctuate due to a wide range of unpredictable meteorological conditions [13].

Several studies have been carried out on prediction technologies for solar irradiance or photovoltaic power generation and other renewable energy sources. Previously, several analysts have used a variety of modeling techniques in an attempt to create an accurate and well-known model in this field by combining various data [14]. However, because of processing costs, some modeling approaches, such as linear or non-linear regression models, may not be suitable for viable complex forecast tasks [15]. Mathematical models cannot be utilized to achieve accurate results since they require a large number of coefficients and sophisticated computations [16]. Numerous research studies have recently been published on the use of traditional machine learning models such as linear, nonlinear, and artificial intelligence models on renewable energy [17]. Qazi *et* al. studied the application of ANNs, which proposed to handle the modeling jobs using a large number of weather features as inputs, resulting in a more accurate and dependable network than other empirical models, as well as greater flexibility [12]. Another study by Voyant *et* al. looked into applying machine learning algorithms to predict solar irradiation [18]. They discovered that machine learning techniques such as nearest neighbor and bootstrap aggregation enhanced performance and accuracy. To anticipate wind energy, Ata used a multi-layered Perceptron network to achieve higher accuracy [19]. Almonacid *et* al. conducted a study on ANNs that were used to forecast the major parameters that affected the performance of outdoor operational solar systems using low and high photovoltaic systems [20]. Y. Wang has used support vector machines (SVM) and non-linear time series to estimate solar intensity [21], however, the SVM-based method achieves a low error rate, and the kernel selection is mainly reliant on experience. To forecast solar intensity, Tang N. used the Least Absolute Shrinkage and Selection Operator (LASSO) and the Single Index Model (SIM) [22]. Simple machine learning approaches such as SVR, ANN, LASSO, and Ridge, cannot give considerable accuracy and cannot meet the needs of operators in circumstances when data is enormous and requires the modeling of the process of trends. As a result, some unique approaches to overcoming these drawbacks have evolved, including deep learning techniques.

As deep learning techniques are becoming increasingly popular due to their ability to direct the relationship in time series data and the ease with which they can be applied. To achieve more efficiency, extra layers are added to the neural network architecture, which is referred to as layering. Numerous DL models have been developed over the last few decades, including Boltzmann machines, Deep Belief Networks (DBNs), and Recurrent Neural Networks (RNNs) [19]. Boltzmann machines are a type of system that learns by making predictions. The RNN, which is a type of neural network, makes use of the sequential nature of incoming data [23]. It is possible to model time-dependent data using RNNs, and these models give great results when applied to time series data. Long-short-term memory (LSTM) is an advanced form of RNN that is capable of retaining information for significantly long time intervals. It is also one of the most extensively used RNN models for time series data forecasting, particularly well

adapted to forecasting challenges associated with PV solar power output. In this work, different machine learning regressors corresponding to Lasso, Ridge, Elastic net, and support vector regressors are applied to the UMass dataset and deep learning techniques such as LSTM, Bi-LSTM and the comparison shows that deep learning techniques provide better results than machine learning techniques in prediction of solar power.

Deep learning approaches are artificial neural networks with a deep architecture capable of processing enormous amounts of data that have outperformed state-of-the-art results in a variety of classification and regression tasks. The foregoing challenges generated by huge scaled data are predicted to be solved using DL algorithms. Hierarchically, these techniques can develop imagination and discriminate features [24], however, DL approaches have been applied to various domains of renewable energy and have been shown to provide higher forecast accuracy for wind and other renewable energies than traditional methods. The time series prediction problem has been addressed by researchers in recent years, with improved performance reported for deep-learning-based neural network approaches. LSTM NN and stacked autoencoders (SAEs) are two deep-learning-based methods that have demonstrated superior performance when compared to some commonly used traditional prediction models. In the traffic speed prediction aspect, Ma *et* al. used both LSTM NN and SAEs for the prediction problem and found superior performance for LSTM NNs than SAEs [25]. Poudel, P., & Jang, B. used LSTM for the prediction of solar power with one input node two hidden layers, and one output node [26]. Patel, H. also applied LSTM and CNN for solar radiation prediction but the model does not converge and does not provide a satisfactory accuracy with a high value of RMSE. However, it has not been determined which type of deep neural network is the most suited model for solar power forecasting. In this paper, it is certain to investigate the different architectures of DL algorithms in solar energy prediction. Specifically, LSTM, Bi-LSTM neural network, and the GRU neural network approaches are implemented for the prediction of solar power prediction. For the optimization of these neural networks, we employ adaptive moment estimation (Adam) and root means square propagation (RMSProp) optimizers.

## 2.1 Solar dataset

The UMass dataset is used to perform the solar radiation forecasts. The solar weather data in this dataset was produced using sensor traces. The Davis Meteorological station in Amherst, Massachusetts, provided the weather data. The information is made up of weather conditions that are captured every five minutes. UV level, Sunlight, Atmospheric Pressure, Rain, Wind and its direction, Dew-point, Humidity, Wind Chill, and Temperature are some of its features. The records from 2006 to 2013 are used for training and testing. Fig 2 describes the visualization of features of the dataset.

The correlation between the features is acquired through the Poython function data_corr() which gives us the correlation matrix in the form of a heatmap shown in Fig 3, representing the higher solar intensity correlation with Temperature, Chill, Humidity, Dewpoint, and Wind.

## 2.2 Features visualization

Fig 4 shows the visual representation of individual features and their variation concerning time in tabular form.

## 2.3 Proposed methodology

First, to implement the machine learning techniques, some pre-requisite processing of data is done to get accurate output, Fig 5 shows the steps required to fulfill before applying machine learning techniques.

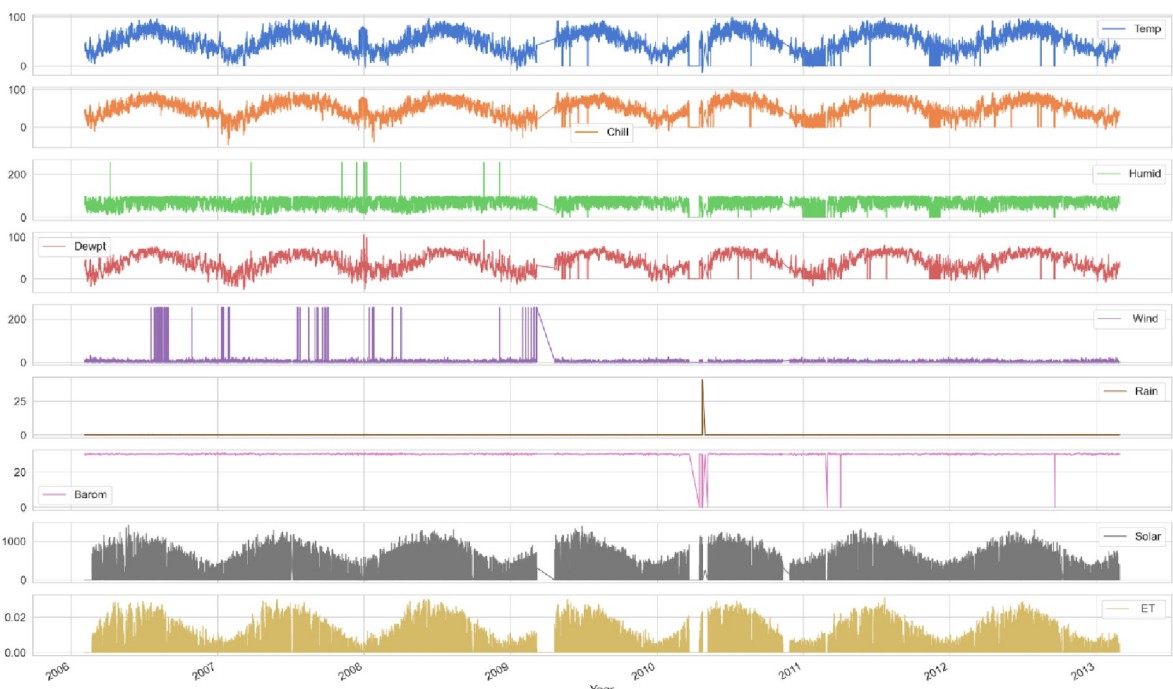

**Fig 2. Data visualization of all features of the solar dataset.**

**2.3.1 Baseline regressors.** At first, Linear regression as a baseline technique is employed. The features with high correlation are preferable to use. Secondly, four machine learning regression techniques are used least absolute shrinkage and selection operator (LASSO), Ridge

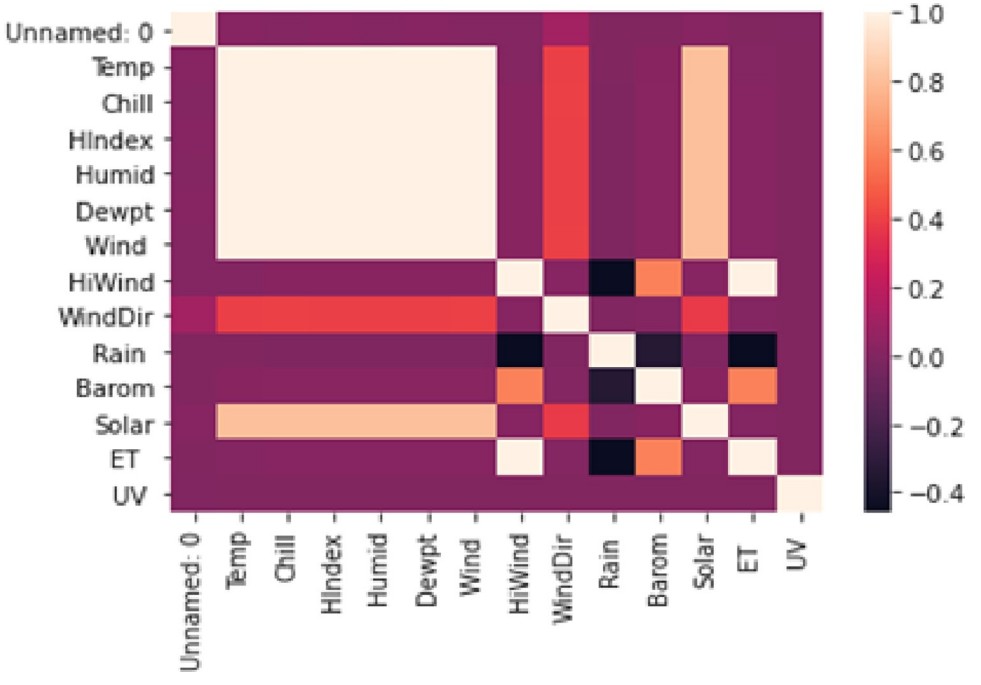

**Fig 3. Heatmap of all solar dataset features.**

| Timestam | Temp | Chill | HIndex | Humid | Dewpt | Wind | HiWind | WindDir | Rain | Barom | Solar | ET |
|---|---|---|---|---|---|---|---|---|---|---|---|---|
| 20060202 | 44.9 | 44.9 | 44.9 | 60 | 31.9 | 3 | 3 | 180 | 0 | 30.071 | 0 | 0 |
| 20060202 | 44.9 | 44.9 | 44.9 | 61 | 32.3 | 1 | 3 | 158 | 0 | 30.075 | 0 | 0 |
| 20060202 | 44.8 | 44.8 | 44.8 | 61 | 32.2 | 2 | 3 | 158 | 0 | 30.075 | 0 | 0 |
| 20060202 | 44.8 | 44.8 | 44.8 | 62 | 32.6 | 1 | 2 | 158 | 0 | 30.075 | 0 | 0 |
| 20060202 | 44.8 | 44.8 | 44.8 | 61 | 32.2 | 2 | 5 | 135 | 0 | 30.074 | 0 | 0 |
| 20060202 | 44.8 | 44.8 | 44.8 | 61 | 32.2 | 3 | 6 | 135 | 0 | 30.074 | 0 | 0 |
| 20060202 | 44.8 | 44.8 | 44.8 | 61 | 32.2 | 3 | 6 | 135 | 0 | 30.074 | 0 | 0 |
| 20060202 | 44.8 | 42.1 | 44.8 | 61 | 32.2 | 5 | 10 | 135 | 0 | 30.066 | 0 | 0 |
| 20060202 | 44.6 | 41.2 | 44.6 | 61 | 32 | 6 | 9 | 135 | 0 | 30.066 | 0 | 0 |
| 20060202 | 44.6 | 42.6 | 44.6 | 61 | 32 | 4 | 7 | 135 | 0 | 30.066 | 0 | 0 |
| 20060202 | 44.5 | 42.5 | 44.5 | 61 | 31.9 | 4 | 6 | 135 | 0 | 30.069 | 0 | 0 |
| 20060202 | 44.5 | 44.5 | 44.5 | 62 | 32.3 | 3 | 7 | 112 | 0 | 30.069 | 0 | 0 |
| 20060202 | 44.3 | 41.5 | 44.3 | 62 | 32.1 | 5 | 9 | 135 | 0 | 30.069 | 0 | 0 |
| 20060202 | 44.3 | 42.3 | 44.3 | 62 | 32.1 | 4 | 7 | 135 | 0 | 30.077 | 0 | 0 |
| 20060202 | 44.3 | 44.3 | 44.3 | 62 | 32.1 | 2 | 4 | 158 | 0 | 30.077 | 0 | 0 |
| 20060202 | 44.2 | 42.2 | 44.2 | 62 | 32 | 4 | 6 | 135 | 0 | 30.077 | 0 | 0 |
| 20060202 | 44.1 | 42.1 | 44.1 | 63 | 32.3 | 4 | 6 | 135 | 0 | 30.082 | 0 | 0 |
| 20060202 | 43.8 | 43.8 | 43.8 | 64 | 32.4 | 2 | 6 | 135 | 0 | 30.082 | 0 | 0 |
| 20060202 | 43.3 | 43.3 | 43.3 | 65 | 32.3 | 1 | 5 | 135 | 0 | 30.082 | 0 | 0 |
| 20060202 | 43.5 | 40.6 | 43.5 | 64 | 32.1 | 5 | 10 | 135 | 0 | 30.087 | 0 | 0 |
| 20060202 | 43.6 | 41.5 | 43.6 | 64 | 32.2 | 4 | 10 | 135 | 0 | 30.087 | 0 | 0 |

**Fig 4. Individual feature visualization.**

Regression, ElasticNet Regression, and Support Vector Regression (SVR). A detailed description of these methods is given in the proceeding section.

*a. LASSO.* Least Absolute Shrinkage and Selection operator. It is used for data regularization and feature selection. LASSO is a regularization technique. Due to its accurate prediction, it is used as a regression model. It uses a shrinkage operator. Shrinkage is modeled in this way. Data values are shrunk towards a central point, which is known as the mean when shrinkage occurs. Simple, sparse models benefit from the lasso method. For models with high levels of multicollinearity or when specific portions of model selection, such as variable selection/parameter removal, need to be automated, this particular sort of regression is well-suited. The error function LASSO is in Eq 1:

$$\sum (y_i - \sum_j x_{ij}\beta_j)^2 + \lambda \sum_{j=1}^{p} |\beta_i| \tag{1}$$

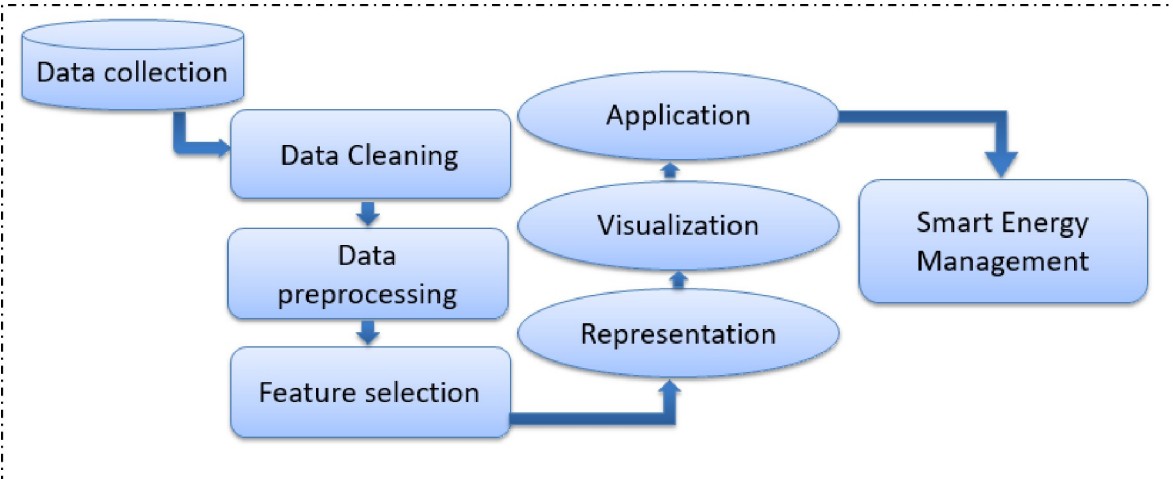

**Fig 5. Workflow diagram of the proposed technique.**

Where, $\lambda$ is the amount of shrinkage and $\beta$ is the penalty term of the L1 Regularization in Lasso regression It is employed when there are a lot of features because it does the feature selection for the model automatically.

*b. Ridge regression.* Ridge regression uses L2 regularization. It is a model tuning method that is used to analyze multicollinearity data, however, when there is a problem of multicollinearity in our data, variances are huge and the predictions are so far from actual values. The error function Ridge regression uses:

$$L_{ridge}(\hat{\beta}) = \sum_{i=1}^{n}(y_i - x_1'\hat{\beta})^2 + \lambda\sum_{j=1}^{m}w_j\hat{\beta}_J^2 \tag{2}$$

Where, $\lambda$ is the penalty term. The $\beta$ parameter in the ridge function denotes the value of the function's argument. We may thus manipulate the penalty term by varying the value of $\beta$. The penalty is greater the higher the $\beta$ value, therefore the coefficients have a smaller magnitude. It reduces the boundaries. Consequently, it's employed to keep multicollinearity at bay. It decreases the model complexity by coefficient shrinking. It penalizes the sum of the square of weights and has a non-sparse solution. It does not select features and gives better predictions when our variable is a function of all input features, but it can learn complex data patterns.

*c. ElasticNet regression.* Before Elastic Net, critics had pointed out that the variable selection in LASSO is overly reliant on data and thus unstable. The best of both worlds can be achieved by combining ridge regression and lasso penalties. Elastic Net tries to reduce the following loss function to the absolute minimum possible:

$$L_{enet}\left(\hat{\beta}\right) = \frac{\sum_{i=1}^{n}(y_i - \acute{x}_i\hat{\beta})^2}{2n} + \lambda\left(\frac{1-\alpha}{2}\sum_{j=1}^{m}\hat{\beta}_j^2 + \alpha\sum_{j=1}^{m}|\hat{\beta}_i|\right) \tag{3}$$

where $\alpha$ is the mixing parameter of Lasso and Ridge because ElasticNet uses both L1 and L2 regularization.

*d. Support vector regressor.* SVR has been used the same method of support vector machine (SVM) therefore employed for regression problems. SVR provides the flexibility to set the amount of error that is acceptable in the proposed model, and it will identify the most suited line to fit the solar data. It is a powerful algorithm that helps to determine how tolerant the models of errors are, both in terms of an acceptable error margin and in terms of tuning our tolerance for slipping outside of that acceptable error rate. Expectantly, these merits have demonstrated the ins and outs of SVR and specified the confidence to use it in different data environments. The objective function of SVR is:

$$MIN\frac{1}{2}||w||^2 + c\sum_{i=1}^{n}|\varepsilon_i| \tag{4}$$

There is another hyperparameter, C, that may modify solar data preferences. Tolerance for points outside of the normal distribution increases as C grows larger. In proportion to the decreasing value of C, the tolerance decreases and the equation decomposes into the simplified (but occasionally infeasible) one.

**2.3.2 Deep recurrent neural networks.** A recurrent neural network is a form of artificial neural network that uses sequential data or time series data to train its neural network models. Different studies are approved that it is difficult to forecast solar PV time series data using solely computational intelligence approaches, such as ANN, because of their dynamic behavior, autoregressive, and weather dependence. They have poor predicting capacity because they are ineffective at identifying the behavior of nonlinear time series. This research employed the

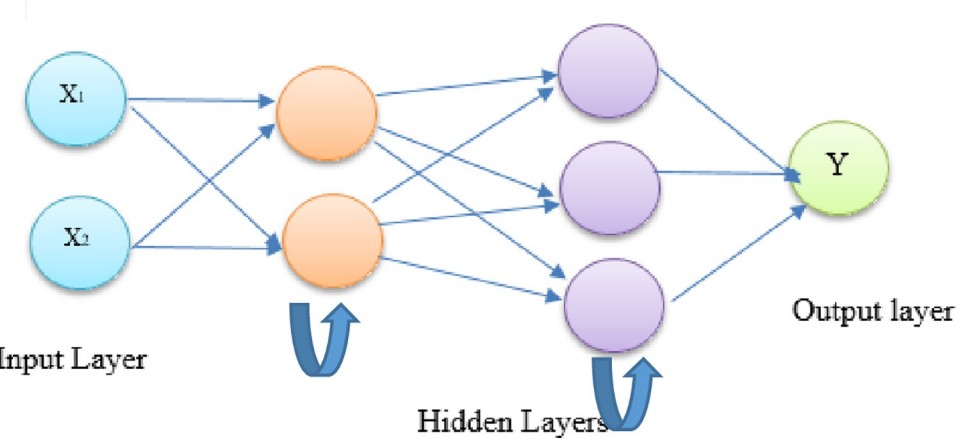

**Fig 6. Layered architecture of RNN.**

LSTM model to extract the weather dependency as well as the regressive component from the data, overcoming these restrictions. To generate more precise prediction results for the forecast of solar PV generation, this work has combined the capacity of LSTM to retain knowledge from the past with the power gates to derive regression rules from meteorological data. The architecture of RNN is presented in Fig 6. Commonly used RNN architectures include LSTM, Bi-LSTM, and GRU, which are discussed below.

*a. LSTM.* Long short-term Memory is a recurrent neural network that is used for time series data prediction because it solves the long-term dependency problem. It is different from other neural networks in a way that it has memory to store short-term data [27]. All recurrent neural networks take the form of a chain of neural network modules that are repeated over and over again. In standard RNNs, this repeating module will have a very simple structure, such as a single *tanh* layer, and will be repeated many times. LSTMs have a chain-like structure as well, but the repeating module has a different structure than the other modules. Instead of having a single neural network layer, there are four of them, each of which interacts in a unique manner [28]. They are extremely effective at solving a wide range of problems, and they are now widely employed. LSTMs are specifically designed to avoid the problem of long-term dependency in the first place. Remembering information for extended periods is practically second nature to them; it is not something they have to work hard to master. Fig 7 defines the model diagram of LSTM.

There are three types of gates in the LSTM Cell which repeat: input gate, output gate, and forget gate. The activation function used in these gates is Sigmoid as it provides the positive values and tells which feature of data needs to be kept and which to be discarded. The input gate describes what information is needed to be stored in the cell state. A cell state is a long-term memory that keeps the necessary information and discards the unnecessary. Input state is a sigmoid function applied to the sum of weights of the previous hidden state and the input of the current cell and bias. Eq (5) shows us how the input state works:

$$i_t = \sigma(w_i[h_{t-1}, x_t] + b_i) \tag{5}$$

Where $i_t$ represents the input gate, $\sigma$ is for the sigmoid function, $x_t$ is the input at the current timestamp, $h_{t-1}$ is the output of the previous LSTM block. While $w_i$ and $b_i$ represent the weight and bias of the input gate.

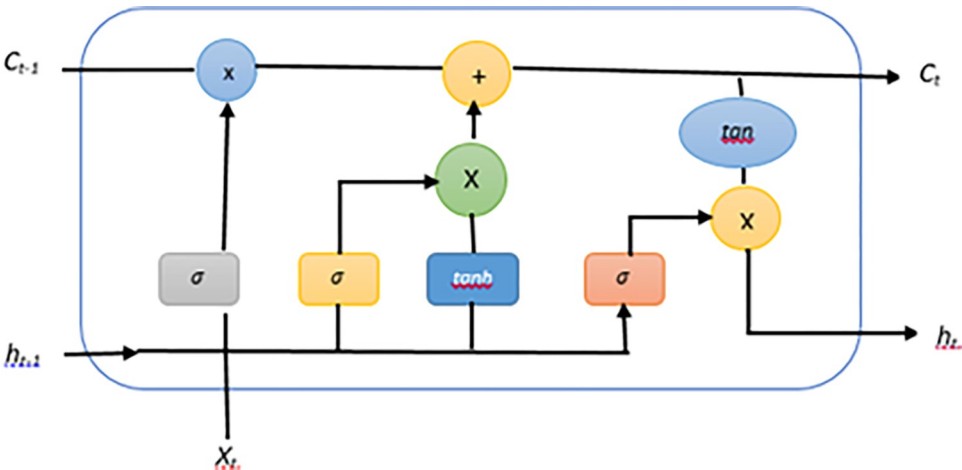

**Fig 7. Single-cell of LSTM.**

The forget gate tells the model which of the historic information to be discarded and which to be kept. Eq 6 describes the mathematical working for forget gate:

$$f_t = \sigma(w_f[h_{t-1}, x_t] + b_i) \tag{6}$$

$h_{t-1}$ is the information from the previous hidden state, $\sigma$ is applied on the input of the current cell with an output of the previous hidden state as input to forget gate. The output gate takes the same inputs as forget gate:

$$o_t = \sigma(w_o[h_{t-1}, x_t] + b_o) \tag{7}$$

Six variants of LSTM have been proposed to evaluate the efficacy of the algorithm.

*b. Bi-LSTM*. Bidirectional recurrent neural nets are constructed by presenting each training sequence both forward and backward to two independent recurrent neural nets, both of which are connected to the same output layer [29]. This means that the BRNN possesses comprehensive, sequential information about every point in a particular sequence, including all points before and after it. There is also no need to determine a (task-dependent) time window or goal delay size because the network can use as much or as little of this context as it needs. After all, it is completely free to do so. As a result, Bi-LSTMs significantly enhance the quantity of information available to the network, hence enhancing the context provided to the algorithm.

*c. Gated recurrent unit*. GRU is the latest generation of Recurrent Neural Networks, and it is quite similar to the LSTM in terms of functionality. They did away with the cell state and instead made use of the hidden state to transport information. In addition, it only contains two gates: a reset gate and an update gate, which are both identical as shown in Fig 8.

$r_t$ shows the output of the reset gate and $z_t$ is used for the update gate, these gates are defined in the proceeding section.

## Reset Gate

The Reset Gate is in charge of the network's short-term memory, which is also known as the "hidden state" ($H_t$). The Reset gate's equation is given.

$$r_t = \sigma(x_t * U_r + H_{t-1} * W_r) \tag{8}$$

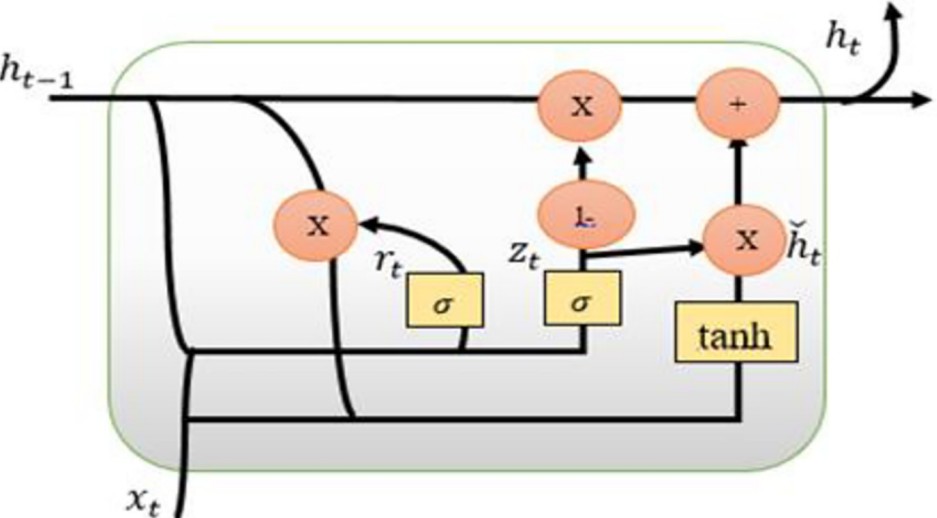

**Fig 8. Illustration of a single cell of GRU.**

## Update Gate

The difference between the reset gate and the update gate is just weights. The equation of the Update gate is:

$$u_t = \sigma(x_t * U_u + H_{t-1} * W_u) \tag{9}$$

## Hidden State

Our current hidden state $H_t$ will be constructed by using the candidate state once we have it. This is where the Update gate comes into play. A fascinating equation has emerged for the first time: in GRU, we utilize a single update gate to govern both the previous data (which comes from the candidate state) and the present data (which comes from the state itself) [30].

$$H_t = u_t . H_{t-1} + (1 - u_t) . \acute{H}_t \tag{10}$$

This means that the new hidden state won't have much information from its previous hidden state because the value of $u_t$ is near zero, as previously indicated. The second component, on the other hand, becomes nearly identical to the first, which effectively indicates that the hidden state at the present timestamp will contain solely the information obtained from the candidate state.

The important hyper-parameters that is the number of hidden neurons, the optimizer, and the number of lags (look back) essential to be decided. Parameter optimization plays a very important role to achieve higher accuracy, better efficiency, and good convergence. We executed the Grad student descent (babysitting AKA) trial and error rule to optimize each hyper-parameter as listed in Table 1, keeping the rest constant, and then selected the best parameter values as a trade-off between accuracy and time consumption. The experimental results for parameter optimization used a different number of Lags, which is achieved by computing the best score of the validation set, based on the score number of lags selected. Fig 11 presented the Learning curves of LSTM-V, Bi-LSTM, and GRU-II techniques with the best performance in terms of selecting the data records for forecasting. The variants are created based on different parameters and the corresponding parameter settings are listed in Table 1.

**Table 1. Parameter settings of variants of RNN deep learning techniques.**

| Proposed Variants | No. of Layers | No. of Neurons | Optimizer | Activation Function | Batch Size | No. of Epochs |
|---|---|---|---|---|---|---|
| LSTM | 11 | 128 | Adam | LeakyRelu (Alpha = 0.5) | 256 | 30 |
| LSTM-I | 11 | 128 | Adam | Relu | 256 | 30 |
| LSTM-II | 11 | 256 | Adam | LeakyRelu (Alpha = 0.5) | 256 | 30 |
| LSTM-III | 20 | 128 | Adam | LeakyRelu (Alpha = 0.5) | 256 | 30 |
| LSTM-IV | 11 | 128 | RMSProp | LeakyRelu (Alpha = 0.5) | 256 | 30 |
| LSTM-V | 11 | 128 | Adam | LeakyRelu (Alpha = 0.1) | 256 | 30 |
| Bi-LSTM | 11 | 128 | Adam | LeakyRelu (Alpha = 0.5) | 256 | 30 |
| Bi-LSTM-I | 11 | 128 | Adam | Relu | 256 | 30 |
| Bi-LSTM-II | 11 | 256 | Adam | LeakyRelu (Alpha = 0.5) | 256 | 30 |
| Bi-LSTM-III | 20 | 128 | Adam | LeakyRelu (Alpha = 0.5) | 256 | 30 |
| Bi-LSTM-IV | 11 | 128 | RMSProp | LeakyRelu (Alpha = 0.5) | 256 | 30 |
| Bi-LSTM-V | 11 | 128 | Adam | LeakyRelu (Alpha = 0.5) | 256 | 30 |
| GRU | 11 | 128 | Adam | LeakyRelu (Alpha = 0.1) | 256 | 30 |
| GRU-I | 11 | 128 | Adam | Relu | 256 | 30 |
| GRU-II | 11 | 256 | Adam | LeakyRelu (Alpha = 0.5) | 256 | 30 |
| GRU-III | 20 | 128 | Adam | LeakyRelu (Alpha = 0.5) | 256 | 30 |
| GRU-IV | 11 | 128 | RMSProp | LeakyRelu (Alpha = 0.5) | 256 | 30 |
| GRU-V | 11 | 128 | Adam | LeakyRelu (Alpha = 0.1) | 256 | 30 |

**2.3.3 Performance measures.** For the evaluation of Solar power prediction, four performance measures based on RMSE, MSE, MAE and MAPE are used.

**RMSE** can be calculated as:

$$RMSE = \sqrt{\frac{\sum_{i=1}^{n} ||Y_i - \hat{Y}_i||^2}{n}} \tag{11}$$

Where $n$ represents the number of data points, $Y_i$ is the *ith* measurement, and $\hat{Y}_i$ denotes its corresponding prediction.

**MSE** stands for mean squared error. It calculates the mean of squared error and it is commonly used for model error measuring.

$$MSE = \frac{1}{n}\sum_{i=1}^{n}(Y_i - \hat{Y}_i)^2 \tag{12}$$

Here, $n$ represents the number of data points, $Y_i$ represents actual values and $\hat{Y}_i$ represents forecast values. MAE stands for mean absolute error. It is the mean of all absolute errors.

$$MAE = \frac{1}{n}\sum_{i=1}^{n}|Y_i - \hat{Y}_i| \tag{13}$$

where $|Y_i - \hat{Y}_i|$ defines the absolute error.

**MAPE** is the mean absolute percentage error, which measures the accuracy of our forecasting system in terms of percentage.

$$MAPE = \frac{1}{n}\sum_{t=1}^{n}|\frac{A_t - F_t}{A_t}| \tag{14}$$

Here $A_t$ and $F_t$ are actual and forecasting values, respectively.

## 3. Experimental results and discussion

The UMass dataset from the Amherst weather station is employed in the present work. Fig 9 demonstrates the overall workflow of the proposed schema. With standard preprocessing techniques, it is prepared as an input to our proposed forecasting models for future predictions. The dataset comprises 12 features with a short interval of 5 minutes for 7,06,635 instances.

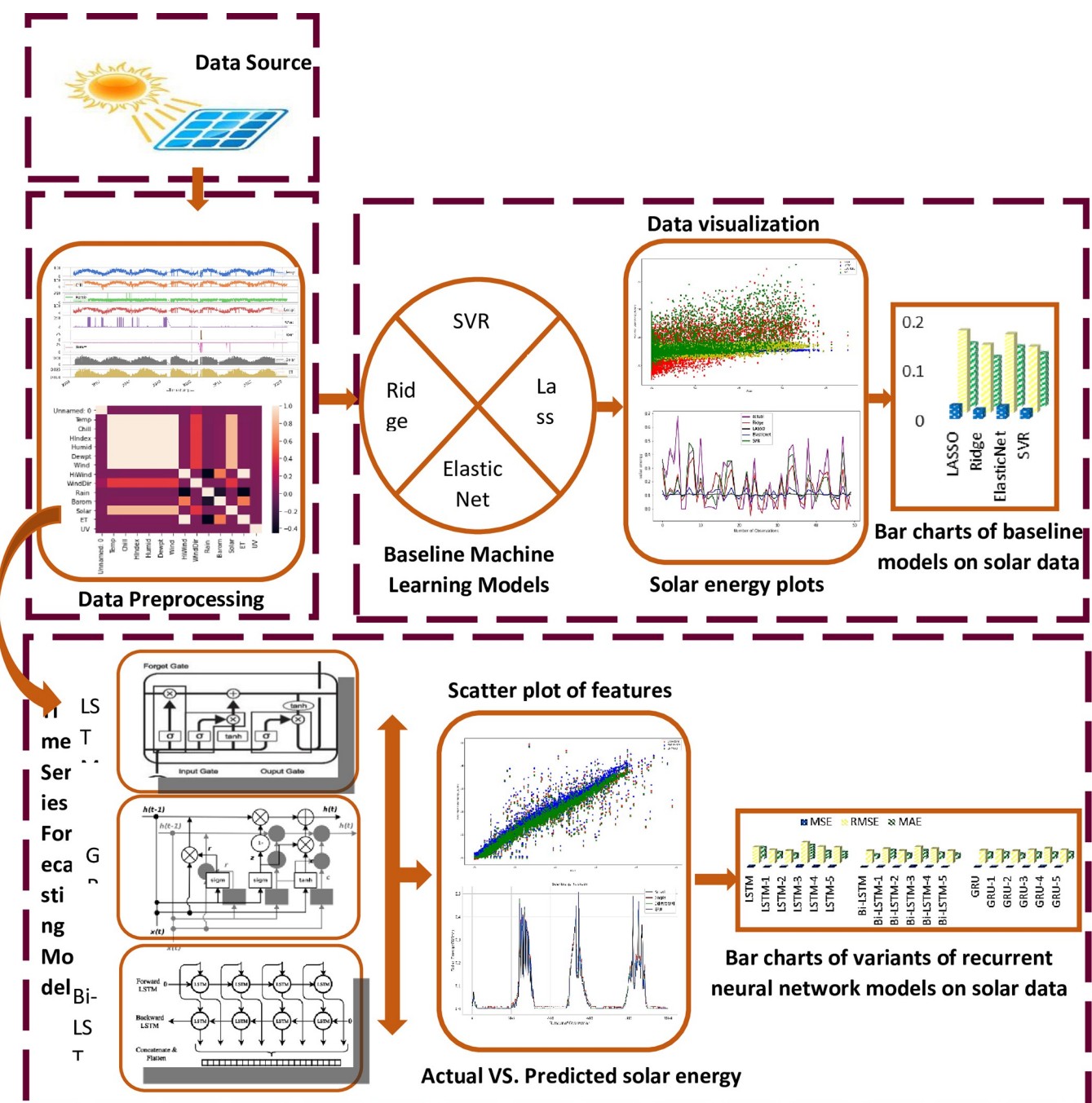

**Fig 9. Representation of the overall workflow schema of the proposed variants of deep time series forecasting techniques.**

Initially, the correlation and interdependency of features are studied through a heat map as demonstrated in Fig 3. It can be observed from this heatmap that features with display Temp, Chill, H-index, Humid, Dewpt, and Wind, show a strong correlation with solar energy, meanwhile, Rain, ET, and Hi-Wind have a over negative correlation. For solar energy prediction, in the first phase baseline machine learning regressors are employed.

LASSO is employed with L1 Regularization factor which helps in the selection of features. Furthermore, Ridge regression is implemented, as LASSO does not provide pretty satisfactory results. Ridge regression predicts more accurately than LASSO because it uses L2 regularization and tunes the data to resolve the problem of multicollinearity which occurs when using LASSO regression.

It can be observed from sub-figures (a & b) of Fig 10 demonstrating actual vs predicted solar energy plots for Lasso and Ridge regression models, respectively. Further, another regression model of same category, ElasticNet regression is implemented which overcomes the limitations of both Lasso and Ridge regression models by use of L1 and L2 regularization as discussed earlier in Section 2.3.1. Subfigure 10(C) demonstrates that Ridge regression model gives better performance than the ElasticNet technique on the UMass dataset. SVR is also implemented and the result can be seen in the form of an energy plot as shown in Fig 10(D). These energy plots show that SVR predicts solar energy with higher accuracy because the predicted values are closer to the original recorded values.

Comparison of these regression models is carried out in terms of performance metrics including MSE, MAE, and RMSE for better understanding on their performance. These results are presented in Table 2. It can be easily noticed SVR provides the lowest error values of 0.1326 and 0.0175 for RMSE, and MSE, respectively, and a comparable result of 0.106 for MAE. Although, the baseline regressors performed well on UMass dataset, however, these techniques do not solve the long-term dependency problem which occurs when our desired output depends upon the inputs that are present in the past and far from the current input. Therefore, to solve that problem, LSTM is applied and variants of LSTM are generated by changing different parameters such as the number of input neurons, optimizer, activation function, number of hidden layers, changing the value of alpha, parameter setting for all proposed variants of LSTM, Bi-LSTM, and GRU (I-V) is illustrated in Table 1. Bi-LSTM is applied to improve the accuracy of prediction because it handles the data more efficiently by traversing the data in forward and backward directions. And also, the variants of Bi-LSTM using the same changes in parameters are created. Furthermore, gated recurrent neural network is used to reduce complexity as demonstrated in Fig 8, where it utilizes a smaller number of gates while preserving the same functionality. The purpose of creating different variants of these models is to enhance the robustness of our models. To evaluate the performance of the proposed methodologies, convergence of models through learning curves of randomly choosing three proposed techniques is represented in Fig 11(A)–11(C). It can be noticed that Bi-LSTM converges very well. Additionally, performance assessment in terms of energy prediction plots (actual vs predicted) of the selected variants along with the comparison of baseline technique is illustrated in Fig 12. It can be realized that predicted energy is in a good pattern with the actual energy, which is demonstrated in subfigures (c,d). Overall trend can be easily observed from these scatter plots.

However, statistical error measures of MAE, MSE, RMSE and MAPE are numerically provided in Table 3 for further evaluation of each model and relative comparison.

Initially, simple base line regression models have been employed by taking inputs of solar data. These results are demonstrated in Table 2 in terms of all error measures. Formerly, all the variants of proposed RNN techniques are employed as explained earlier. Their corresponding results in terms of performance indices are illustrated in Table 3. It can be observed that

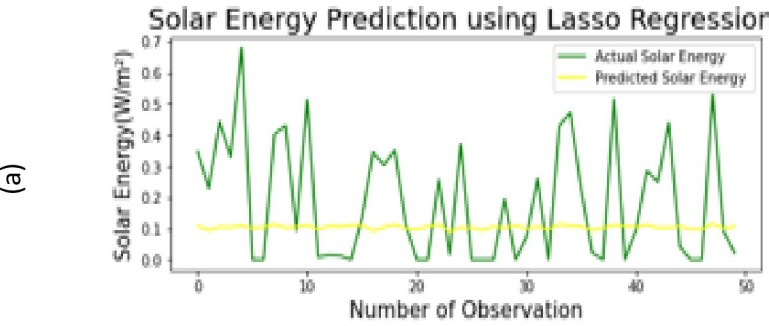

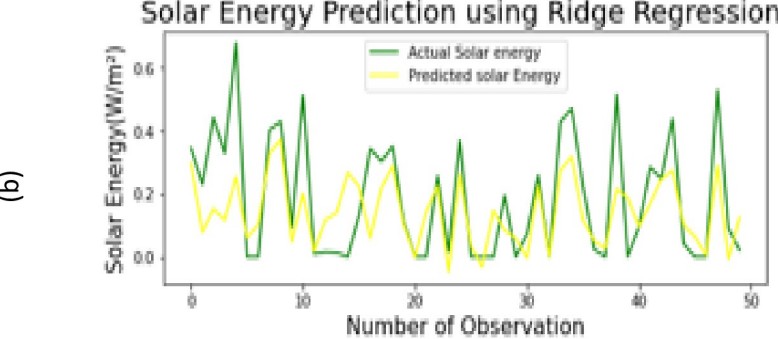

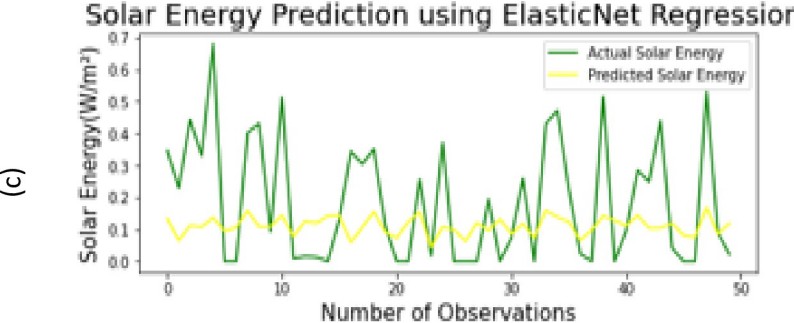

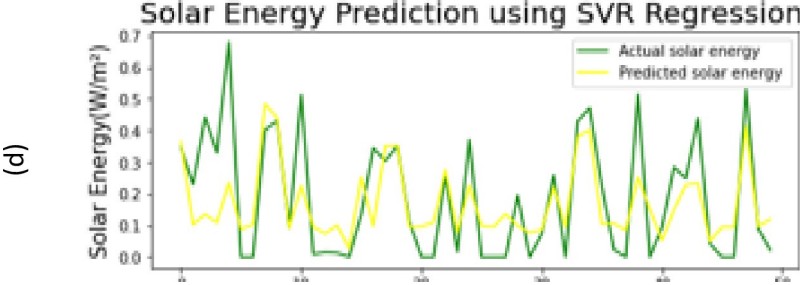

**Fig 10.** Visualization plots (a-d) of actual Vs. predicted solar energy of all baseline regressor.

**Table 2. Error metrics of baseline regressor.**

| Technique | RMSE | MSE | MAE |
|---|---|---|---|
| LASSO | 0.16491 | 0.02719 | 0.12676 |
| Ridge | 0.13634 | 0.01858 | 0.09923 |
| Elastic Net | 0.15761 | 0.02484 | 0.12022 |
| SVR | 0.13260 | 0.01758 | 0.10672 |

among variants of each technique, Bi-LSTM-V and GRU-II shows the lowest MSE value of 0.0012. Bi-LSTM has the MAE value of 0.0124 and GRU-II represents the less error in terms of RMSE with the value of 0.0354. Although RMSE results are comparable with the Bi-LSTM which shows the 0.0356 RMSE value. Additionally, the MAPE values demonstrate the Bi-LSTM-V is good forecasting model with less than 20% value. Through these results, it can be concluded that Bi-LSTM variants give more accurate results over the other proposed variants.

To check the model generalization ability, error analysis is shown in form of bar charts of all suggested variants and is illustrated in Fig 13. Through which the sub-Figures (a-d) is evaluated individually among their variants with respect to MAE, MSE, RMSE, and MAPE. It can be noticed that LSTM-II shows the less error in form of MSE, MAE among all other its variants.

On the other hand, Bi-LSTM-II and Bi-LSTM-V shows the comparable and better results as compared to its variants. Furthermore, GRU-II furnishes the lowest values of all error metrics as opposed to all its variants techniques. Our proposed variants of RNN models, Bi-LSTM has demonstrated the best error measures in terms of all error metrics for the UMass dataset of solar energy. To further estimate the performance of the Bi-LSTM among the other two models. Fig 14 represents the comparison bar graphs of Bi-LSTM with LSTM and GRU, separately each variant in terms of MAE, MSE and RMSE, as shown in left column of Fig 14 and in form of MAPE in the right column. It can be noticed that among these proposed variants techniques, Bi-LSTM and GRU have the lowest values of all error measures and provide comparable results over LSTM. Moreover, the value of MAPE shows the best fit model for Bi-LSTM-V at 12.050. and GRU-II at 12.564. However, it can be noticed that due to various fluctuating nature of features and other characteristics of solar stations, our proposed variants techniques performed very well on this dataset.

Additionally, a comparison analysis between the baseline regressors and the proposed variants models is evaluated on test data. It is demonstrated in the form of bar charts with respect to MSE, MAE, and RMSE in Fig 15. The figure illustrates better performance of SVR among baseline regression models. As opposed to these base line regressors, LSTM-II, Bi-LSTM, Bi-

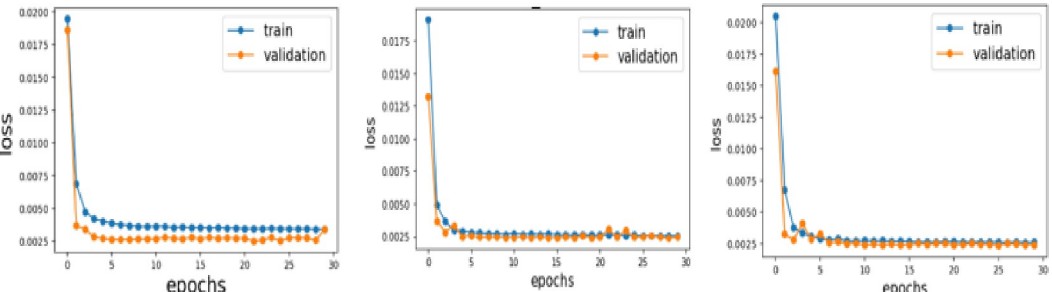

**Fig 11. Learning curves of LSTM-V, Bi-LSTM, and GRU-II techniques with best performance.** a) LSTM-V, b) Bi-LSTM c) GRU- II.

a)  Solar energy prediction using SVR

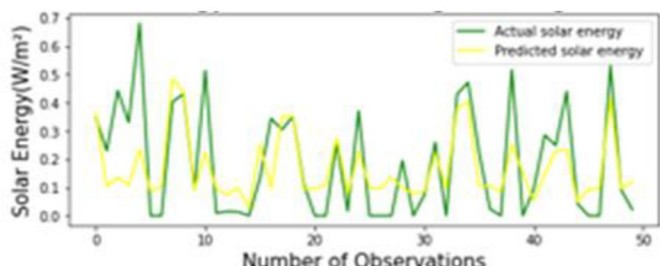

b)  Solar energy prediction using LSTM-V

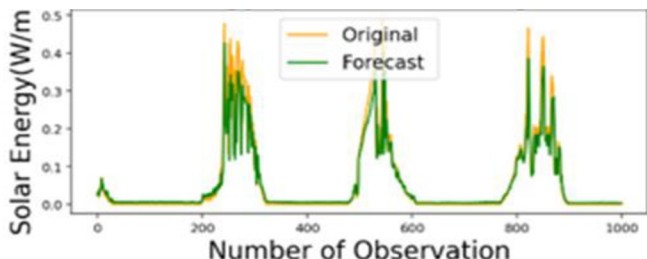

c)  Solar energy prediction using Bi-LSTM

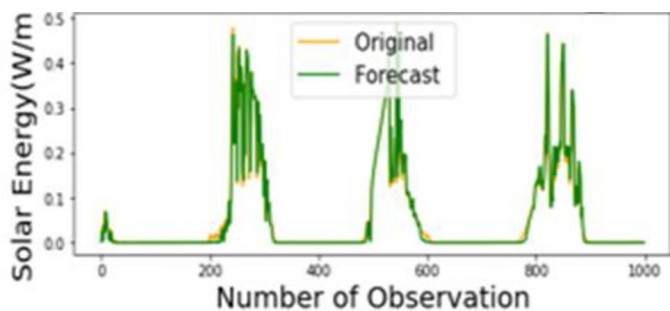

d)  Solar energy prediction using GRU-II

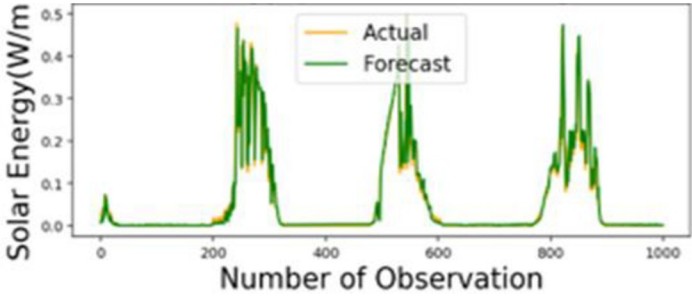

**Fig 12. Solar energy prediction plots of the best performing proposed machine learning and deep learning techniques.** a) Solar energy prediction using SVR, b) Solar energy prediction using LSTM-V, c) Solar energy prediction using Bi-LSTM, d) Solar energy prediction using GRU-II.

**Table 3. Performance measures of the proposed methods.**

| Technique | Variant | MSE | MAE | RMSE | MAPE |
|---|---|---|---|---|---|
| LSTM | LSTM | 0.0022 | 0.0339 | 0.0470 | 54.716 |
| | LSTM-I | 0.0014 | 0.0193 | 0.0386 | 18.603 |
| | LSTM-II | 0.0013 | 0.0140 | 0.0363 | 13.617 |
| | LSTM-III | 0.0036 | 0.0405 | 0.0607 | 61.053 |
| | LSTM-IV | 0.0022 | 0.0214 | 0.0478 | 14.709 |
| | LSTM-V | 0.0021 | 0.0202 | 0.0460 | 14.465 |
| Bi-LSTM | Bi-LSTM | 0.0012 | 0.0124 | 0.0356 | 12.275 |
| | Bi-LSTM-I | 0.0017 | 0.0254 | 0.0415 | 36.197 |
| | Bi-LSTM-II | 0.0013 | 0.0138 | 0.0362 | 12.777 |
| | Bi-LSTM-III | 0.0022 | 0.0248 | 0.0476 | 23.371 |
| | Bi-LSTM-IV | 0.0017 | 0.0169 | 0.0419 | 12.810 |
| | Bi-LSTM-V | 0.0012 | 0.0135 | 0.0359 | 12.050 |
| GRU | GRU | 0.0014 | 0.0179 | 0.0374 | 20.194 |
| | GRU-I | 0.0014 | 0.0187 | 0.0382 | 20.239 |
| | GRU-II | 0.0012 | 0.0138 | 0.0354 | 12.564 |
| | GRU-III | 0.0014 | 0.0179 | 0.0386 | 17.295 |
| | GRU-IV | 0.0018 | 0.0178 | 0.0431 | 12.908 |
| | GRU-V | 0.0014 | 0.0193 | 0.0385 | 19.698 |

LSTM-II, and GRU-II depict the best performance measures between and among all the other proposed variants of time series models. Overall, it can be observed that some improvements with Bi-LSTM is due to its obvious advantage of keeping information preserved for past and future.

To further analyze time series deep learning proposed methodologies, predicted and actual solar energy is statically compared in form of scatter plot, as shown in Fig 16. It demonstrates the linear relationship among all values. A comparison is taken among the existing techniques [24] and the proposed DL time series techniques, and ML baseline methodologies that is illustrated in the Table 4. Here * is for presenting the existing techniques while excluding 55th day. The RMSE, MAPE metric values from the table shows the lower value over the proposed deep learning methods LSTM-II, Bi-LSTM-II, and GRU-II. On the other hand, a comparison with all the baseline and time series techniques with the existing methods is shown in Fig 17 in the form of line graph that describes the huge difference of performance over the previously used methodologies.

A further study is carried out by employing hybrids of LSTM, Bi-LSTM and GRU in six combinations to predict solar energy for time steps of 1, 3 (15 minutes), 6 (half-an-hour) and 12 (an hour) ahead. The results are listed in Table 5 for all five performance indices. It can be observed from their values that these models consistently give a fairly comparable prediction error and endorse the reliability of these models along with robustness. It can be observed that almost all error metrics degrade with increasing time stamps. However, there may be a few violations owing to the entirely stochastic nature of weather. These are further accounted for the reason that we cannot reduce error further by increasing the previous timespan as it may be cloudy or sunny at that time. It can further be assessed that hybrid RNNs produced similar results as those of a single RNN model. The slight difference in various performance indices may be due to the optimization of hyperparameters for each model.

Prediction intervals (PIs) have been calculated for our best-performing algorithm, LSTM_BI-LSTM by exploiting root mean squared forecasting error (RMSFE). The prediction

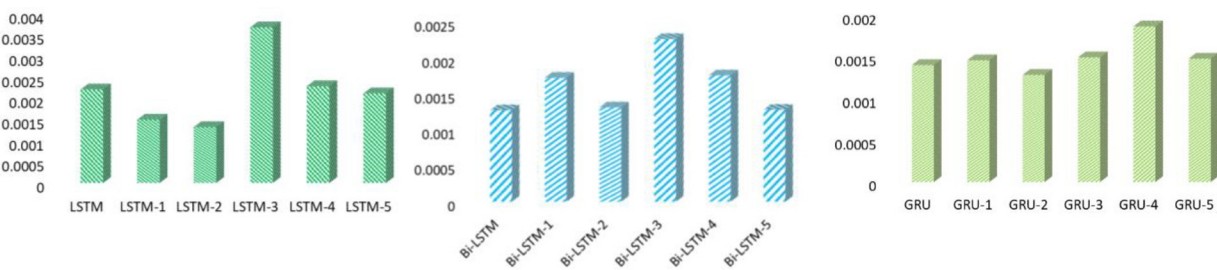

a.  Error analysis of MSE for LSTM, Bi-LSTM, and GRU with its variants

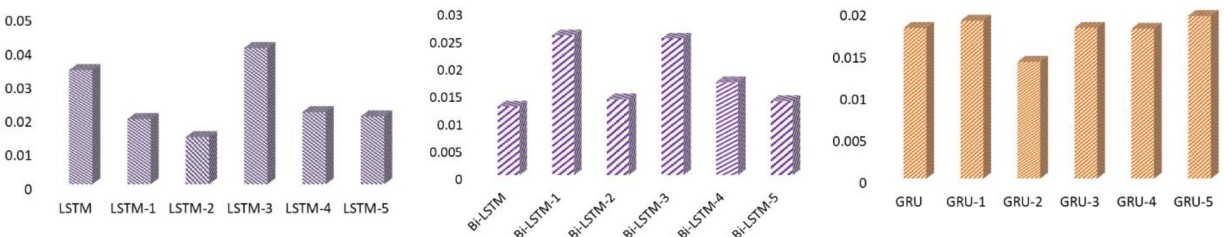

b.  Error analysis of MAE for LSTM, Bi-LSTM, and GRU with its variants

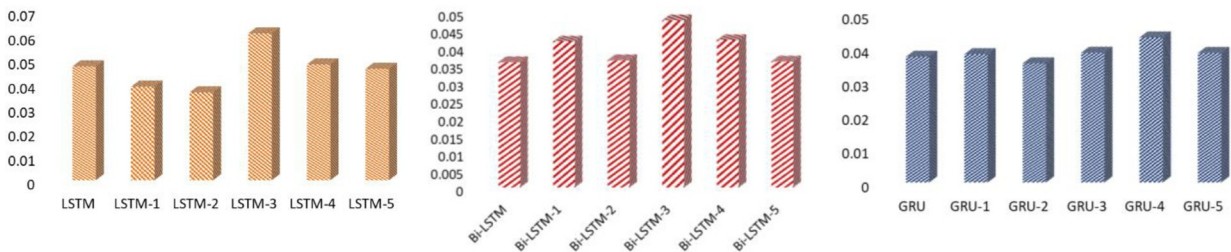

c.  Error analysis of RMSE for LSTM, Bi-LSTM, and GRU with its variants

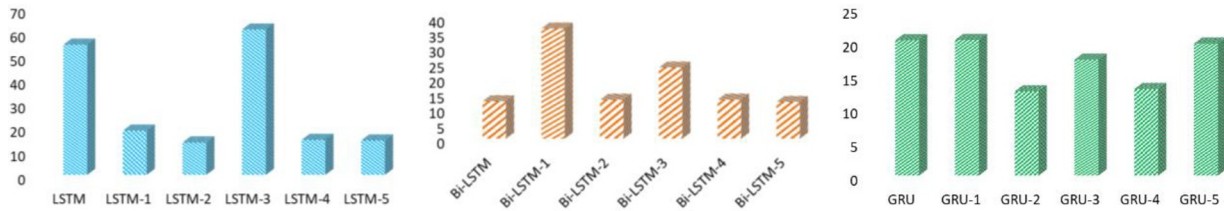

d.  Error analysis of MAPE for LSTM, Bi-LSTM, and GRU with its variants

**Fig 13.** Performance analysis of proposed models (LSTM, Bi-LSTM, and GRU) w.r.t four error metrics MSE, MAE RMSE, and MAPE (a-d). a) Error analysis of MSE for LSTM, Bi-LSTM, and GRU with its variants, b) Error analysis of MAE for LSTM, Bi-LSTM, and GRU with its variants, c) Error analysis of RMSE for LSTM, Bi-LSTM, and GRU with its variants, d) Error analysis of MAPE for LSTM, Bi-LSTM, and GRU with its variants.

interval for the test set at a 95% confidence level is [-0.00213071 0.44163999], and the result is displayed in Fig 18. It can be observed that the predicted values fall well within this interval. The RMSFE method to calculate the prediction interval is very similar to RMSE but calculations for residual errors are made from predictions on unseen data.

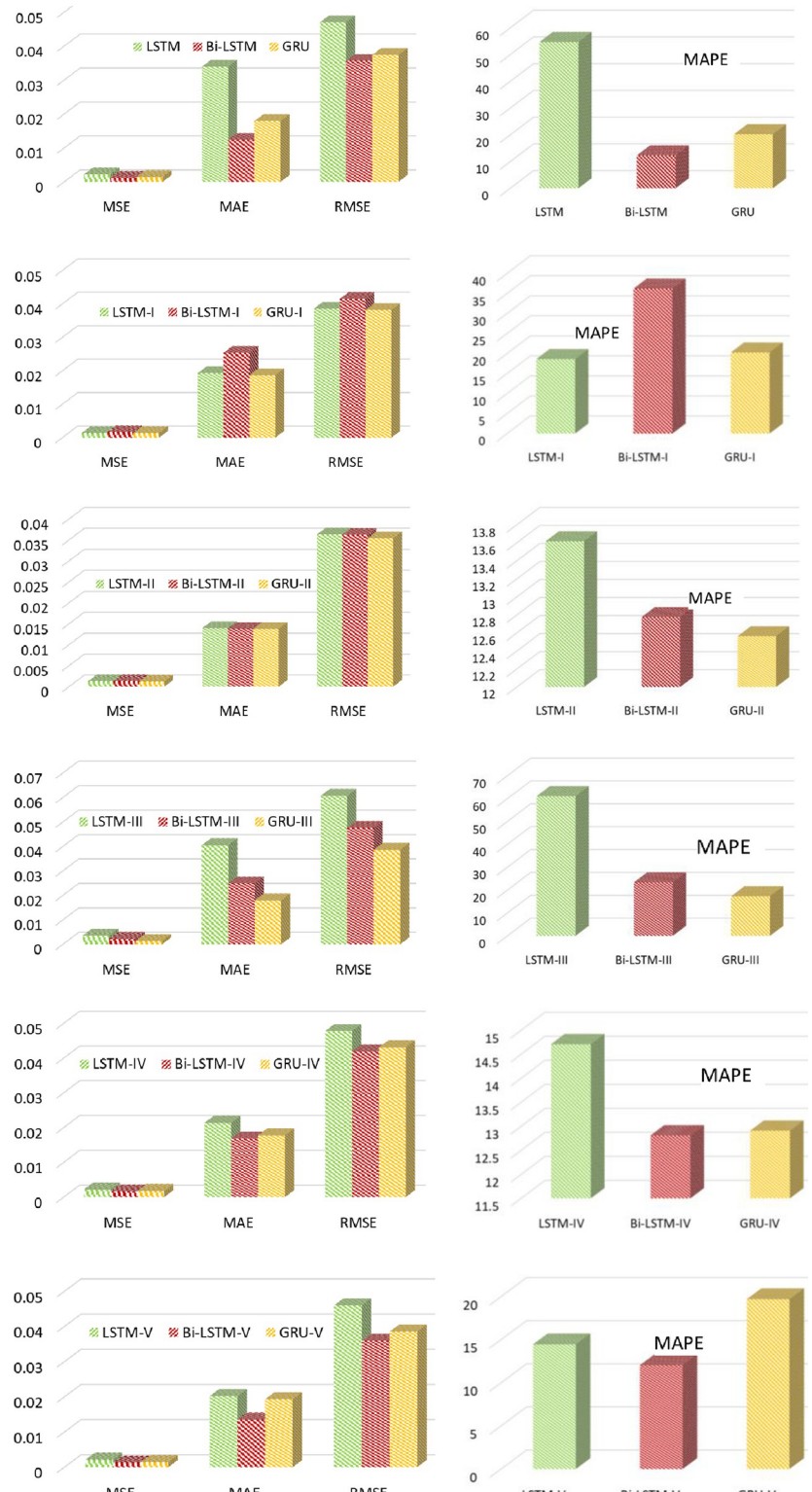

**Fig 14.** Comparison between proposed methodologies (LSTM, Bi-LSTM, and GRU) in terms of MAE, MSE, and RMSE (a, c, e, g, i, k) and MAPE (b, d, f, h, j, l).

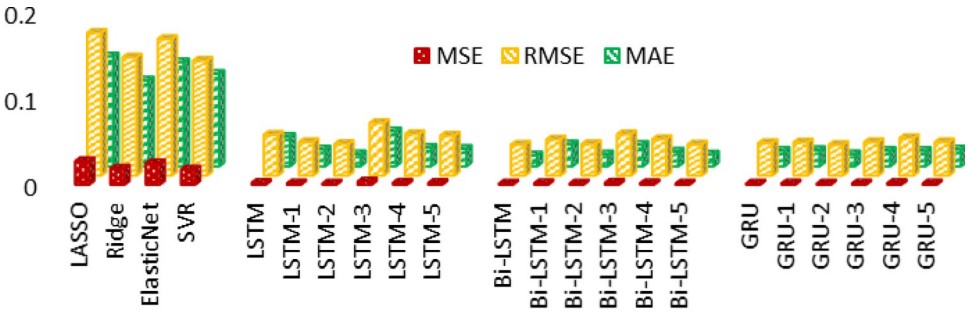

**Fig 15. Comparison between LASSO, Ridge, ElasticNet, and SVR regression models with the proposed variants of deep learning models.**

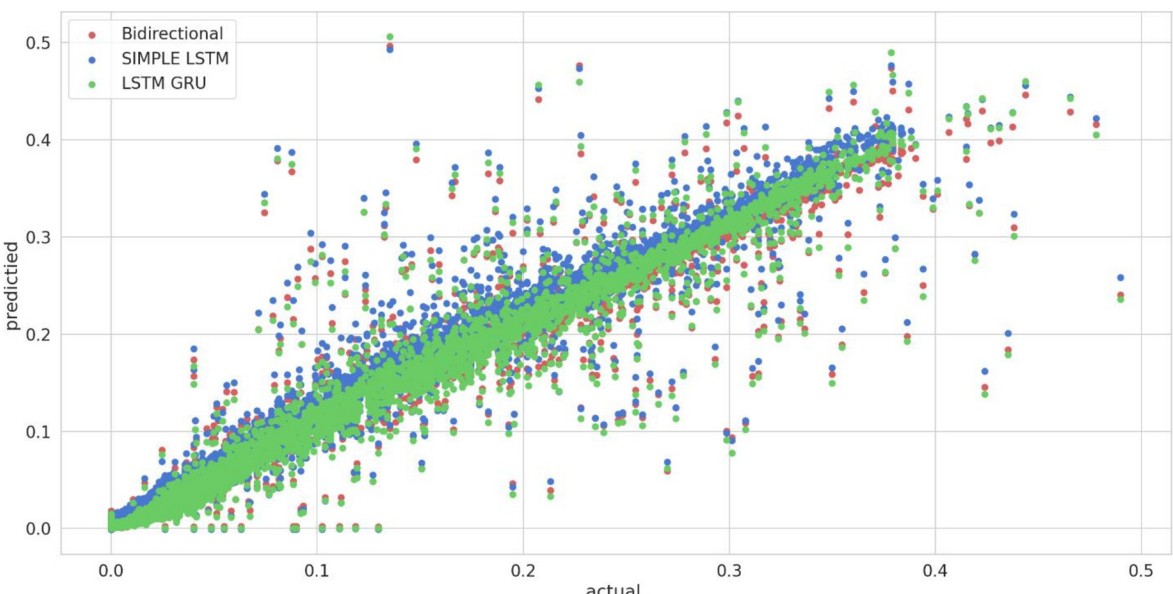

**Fig 16. Predicted vs actual solar energy for deep learning time series models.**

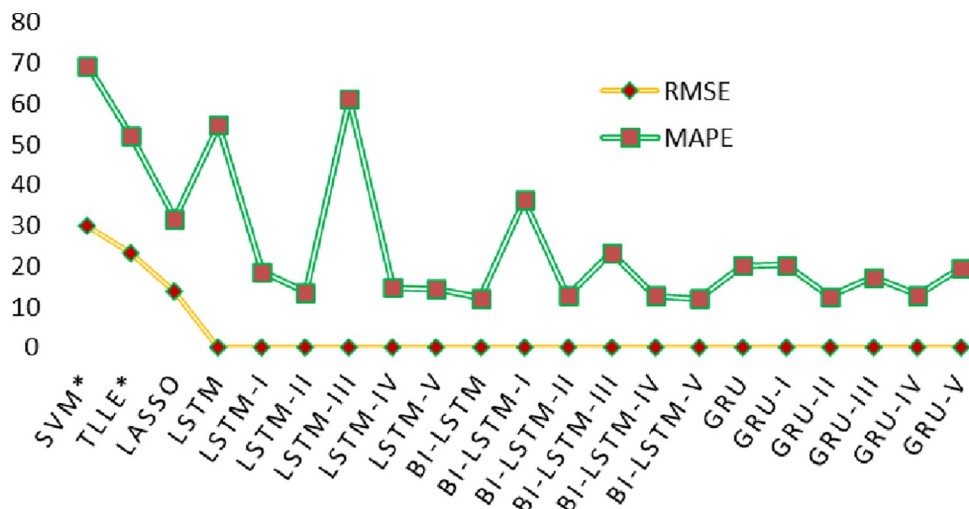

**Fig 17. Line chart comparison between existing techniques [24] and applied ML methods and proposed DL techniques based on RMSE, and MAPE.**

**Table 4. Comparison with reported work in terms of performance indices.**

| Technique | RMSE | MAPE |
|---|---|---|
| SVM* | 30.1524 | 39.2063 |
| TLLE* | 23.1464 | 29.0174 |
| Lasso | 14.0262 | 17.817 |
| LSTM-II | 0.0363 | 13.617 |
| Bi-LSTM-II | 0.0362 | 12.777 |
| GRU-II | 0.0354 | 12.564 |

**Table 5. Performance indices of hybrid RNN models for multistep ahead solar energy prediction.**

| MODEL | MSE | MAE | RMSE | R2SCORE | EVS |
|---|---|---|---|---|---|
| **TIME STEP = 1** | | | | | |
| GRU | 0.00206405 | 0.0220202 | 0.0454318 | 0.887204 | 0.891428 |
| LSTM | 0.0023703 | 0.0242242 | 0.0486857 | 0.880643 | 0.889718 |
| Bi-LSTM | 0.00206113 | 0.0233701 | 0.0453997 | 0.878433 | 0.879702 |
| LSTM_GRU | 0.00185696 | 0.0259492 | 0.0430925 | 0.915883 | 0.926071 |
| LSTM_Bi-LSTM | 0.00333502 | 0.0372468 | 0.0577497 | 0.772848 | 0.779888 |
| Bi-LSTM_LSTM | 0.00481819 | 0.0582489 | 0.0694132 | 0.777946 | 0.905748 |
| Bi-LSTM_GRU | 0.00303340 | 0.0269898 | 0.0550763 | 0.831216 | 0.847649 |
| GRU_LSTM | 0.00347829 | 0.0339021 | 0.0589772 | 0.871959 | 0.891412 |
| GRU_Bi-LSTM | 0.00298206 | 0.0292867 | 0.0546082 | 0.879334 | 0.882642 |
| **TIME STEP = 3** | | | | | |
| GRU | 0.00247863 | 0.0261552 | 0.0497859 | 0.815184 | 0.824407 |
| LSTM | 0.00275324 | 0.0300704 | 0.0524713 | 0.802423 | 0.818601 |
| Bi-LSTM | 0.00302039 | 0.0267444 | 0.0549581 | 0.784867 | 0.811434 |
| LSTM_GRU | 0.00266766 | 0.0280624 | 0.0516494 | 0.794529 | 0.801016 |
| LSTM_Bi-LSTM | 0.00165143 | 0.0202568 | 0.0406378 | 0.910881 | 0.911984 |
| Bi-LSTM_LSTM | 0.00244080 | 0.0255941 | 0.0494044 | 0.834184 | 0.844152 |
| Bi-LSTM_GRU | 0.00233109 | 0.0266523 | 0.0482813 | 0.838331 | 0.840168 |
| GRU_LSTM | 0.00198261 | 0.0206538 | 0.0445263 | 0.885783 | 0.893628 |
| GRU_Bi-LSTM | 0.00208458 | 0.0219832 | 0.0456572 | 0.881187 | 0.890276 |
| **TIME STEP = 6** | | | | | |
| GRU | 0.00190743 | 0.0242073 | 0.0436742 | 0.906045 | 0.913277 |
| LSTM | 0.00290684 | 0.0287056 | 0.0539151 | 0.783197 | 0.805793 |
| Bi-LSTM | 0.00268654 | 0.0253729 | 0.0518318 | 0.813567 | 0.820473 |
| LSTM_GRU | 0.00230535 | 0.0230252 | 0.0480141 | 0.847688 | 0.869296 |
| LSTM_Bi-LSTM | 0.00189955 | 0.0208557 | 0.0435838 | 0.886329 | 0.893798 |
| Bi-LSTM_LSTM | 0.00268097 | 0.0260908 | 0.0517781 | 0.799133 | 0.816882 |
| Bi-LSTM_GRU | 0.00176848 | 0.0220565 | 0.0420533 | 0.896506 | 0.897580 |
| GRU_LSTM | 0.00278891 | 0.0271058 | 0.0528101 | 0.880633 | 0.880683 |
| GRU_Bi-LSTM | 0.00189743 | 0.0213037 | 0.0435596 | 0.901736 | 0.906807 |
| **TIME STEP = 12** | | | | | |
| GRU | 0.00372953 | 0.0352264 | 0.0610699 | 0.796373 | 0.797137 |
| LSTM | 0.00224439 | 0.0237899 | 0.047375 | 0.858809 | 0.871918 |
| Bi-LSTM | 0.0026115 | 0.0235673 | 0.0511028 | 0.823483 | 0.84279 |
| LSTM_GRU | 0.00289865 | 0.0266434 | 0.0538391 | 0.791954 | 0.821393 |
| LSTM_Bi-LSTM | 0.0018357 | 0.0221139 | 0.042845 | 0.891672 | 0.897446 |
| Bi-LSTM_LSTM | 0.00280618 | 0.0264621 | 0.0529734 | 0.829897 | 0.850486 |

*(Continued)*

**Table 5.** (Continued)

| MODEL | MSE | MAE | RMSE | R2SCORE | EVS |
|---|---|---|---|---|---|
| Bi-LSTM_GRU | 0.0033056 | 0.0292154 | 0.0574944 | 0.723444 | 0.748526 |
| GRU_LSTM | 0.00324424 | 0.0296865 | 0.0569582 | 0.851368 | 0.854247 |
| GRU_Bi-LSTM | 0.0030524 | 0.0306670 | 0.0552485 | 0.853821 | 0.854814 |
| TIME STEP = 24 | | | | | |
| GRU | 0.00425422 | 0.0353442 | 0.0652244 | 0.704385 | 0.720188 |
| LSTM | 0.00245289 | 0.025098 | 0.0495266 | 0.843318 | 0.861182 |
| Bi-LSTM | 0.00244217 | 0.022773 | 0.0494183 | 0.848881 | 0.8578 |
| LSTM_GRU | 0.00274225 | 0.0289594 | 0.0523665 | 0.841052 | 0.8578 |
| LSTM_Bi-LSTM | 0.00163621 | 0.0182187 | 0.04045 | 0.920994 | 0.921847 |
| Bi-LSTM_LSTM | 0.00370454 | 0.0310266 | 0.060865 | 0.705887 | 0.754084 |
| Bi-LSTM_GRU | 0.00182356 | 0.0230117 | 0.0427032 | 0.89818 | 0.898611 |
| GRU_LSTM | 0.00283878 | 0.0284374 | 0.0532802 | 0.856582 | 0.857921 |
| GRU_Bi-LSTM | 0.00265936 | 0.0270759 | 0.051569 | 0.855943 | 0.862133 |

## 4. Conclusion

Deep learning-based time series forecasting techniques have been employed to predict accurate solar energy for a very short term on the dataset from a weather station in Amherst USA. Different machine learning techniques such as LASSO, Ridge Regression, Elastic Net Regression, and Support Vector Regression are applied to solar energy prediction. SVR provides the best results among all with RMSE equal to 0.1320, MSE equal to 0.01758 and MAE equal to 0.1067. RNN-based deep learning techniques including LSTM, Bi-LSTM and GRU along with their variants are implemented to get energy predictions. A comparison of error measures demonstrates that variants of Bi-LSTM and GRU furnish better performance measures. Performance metrics RMSE, MSE, and MAE for Bi-LSTM are 0.0356, 0.0012, and 0.0124 and for GRU these values are 0.0354, 0.0012, and 0.0138, respectively. While value of MAPE is 12.2% for Bi-LSTM and 12.5% for GRU.

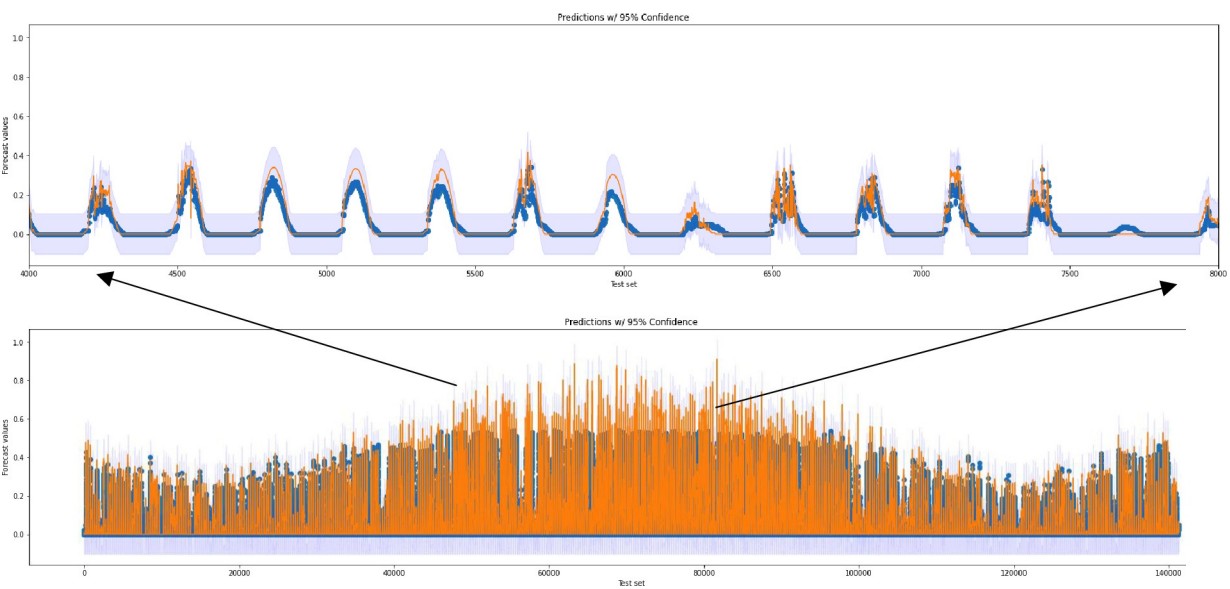

**Fig 18. Prediction intervals on test data of forecast values.**

The consistent performance of variants of deep forecasting models reflects the robust nature of the models with good accuracy and lower prediction error as compared to other machine learning models. Comparison with existing techniques on this dataset in terms of performance indices reveals an enhanced performance of the proposed techniques for almost all its variants. Hybrids of LSTM, Bi-LSTM, and GRU have been employed for multistep ahead prediction of solar energy. The results in terms of performance indices demonstrate the efficacy, reliability, and robustness of the models. It is concluded that LSTMs and GRU can be exploited for reliable, robust, and accurate solar energy prediction.

## Author Contributions

**Conceptualization:** Aneela Zameer.

**Data curation:** Aneela Zameer, Fatima Jaffar.

**Funding acquisition:** Muhammad Muneeb.

**Investigation:** Fatima Jaffar, Farah Shahid, Muhammad Muneeb.

**Methodology:** Aneela Zameer, Fatima Jaffar, Farah Shahid, Muhammad Muneeb.

**Project administration:** Aneela Zameer.

**Resources:** Aneela Zameer.

**Validation:** Farah Shahid, Rizwan Khan.

**Visualization:** Farah Shahid, Rizwan Khan, Rubina Nasir.

**Writing – original draft:** Fatima Jaffar.

**Writing – review & editing:** Aneela Zameer, Rizwan Khan, Rubina Nasir.

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
