## [Decision Letter · Decision Letter 0]

7 Oct 2022

PONE-D-22-24267Short- Short term solar energy forecasting: Integrated computational intelligence of LSTMs and GRUPLOS ONE

Dear Dr. Zameer,

Thank you for submitting your manuscript to PLOS ONE. After careful consideration, we feel that it has merit but does not fully meet PLOS ONE’s publication criteria as it currently stands. Therefore, we invite you to submit a revised version of the manuscript that addresses the points raised during the review process. Please submit your revised manuscript by Nov 18 2022 11:59PM. If you will need more time than this to complete your revisions, please reply to this message or contact the journal office at plosone@plos.org. Please include the following items when submitting your revised manuscript:A rebuttal letter that responds to each point raised by the academic editor and reviewer(s). You should upload this letter as a separate file labeled 'Response to Reviewers'.A marked-up copy of your manuscript that highlights changes made to the original version. You should upload this as a separate file labeled 'Revised Manuscript with Track Changes'An unmarked version of your revised paper without tracked changes. You should upload this as a separate file labeled 'Manuscript'.If applicable, we recommend that you deposit your laboratory protocols in protocols.io to enhance the reproducibility of your results. Protocols.io assigns your protocol its own identifier (DOI) so that it can be cited independently in the future. For instructions see: https://journals.plos.org/plosone/s/submission-guidelines#loc-laboratory-protocols. Additionally, PLOS ONE offers an option for publishing peer-reviewed Lab Protocol articles, which describe protocols hosted on protocols.io. Read more information on sharing protocols at https://plos.org/protocols?utm_medium=editorial-email&utm_source=authorletters&utm_campaign=protocols.

We look forward to receiving your revised manuscript. Kind regards, Shuo-Yan Chou

Academic Editor

PLOS ONE Journal Requirements: When submitting your revision, we need you to address these additional requirements 1. Please ensure that your manuscript meets PLOS ONE's style requirements, including those for file naming. The PLOS ONE style templates can be found at 

https://journals.plos.org/plosone/s/file?id=ba62/PLOSOne_formatting_sample_title_authors_affiliations.pd 2. We suggest you thoroughly copyedit your manuscript for language usage, spelling, and grammar. If you do not know anyone who can help you do this, you may wish to consider employing a professional scientific editing service. Whilst you may use any professional scientific editing service of your choice, PLOS has partnered with both American Journal Experts (AJE) and Editage to provide discounted services to PLOS authors. Both organizations have experience helping authors meet PLOS guidelines and can provide language editing, translation, manuscript formatting, and figure formatting to ensure your manuscript meets our submission guidelines. To take advantage of our partnership with AJE, visit the AJE website (http://learn.aje.com/plos/) for a 15% discount off AJE services. To take advantage of our partnership with Editage, visit the Editage website (www.editage.com) and enter referral code PLOSEDIT for a 15% discount off Editage services.  If the PLOS editorial team finds any language issues in text that either AJE or Editage has edited, the service provider will re-edit the text for free. Upon resubmission, please provide the following: a. The name of the colleague or the details of the professional service that edited your manuscript.

b. A copy of your manuscript showing your changes by either highlighting them or using track changes (uploaded as a *supporting information* file). 

c. A clean copy of the edited manuscript (uploaded as the new *manuscript* file) 3. Please note that PLOS ONE has specific guidelines on code sharing for submissions in which author-generated code underpins the findings in the manuscript. In these cases, all author-generated code must be made available without restrictions upon publication of the work. Please review our guidelines at https://journals.plos.org/plosone/s/materials-and-software-sharing#loc-sharing-code and ensure that your code is shared in a way that follows best practice and facilitates reproducibility and reuse. 4. We note that you have stated that you will provide repository information for your data at acceptance. Should your manuscript be accepted for publication, we will hold it until you provide the relevant accession numbers or DOIs necessary to access your data. If you wish to make changes to your Data Availability statement, please describe these changes in your cover letter and we will update your Data Availability statement to reflect the information you provide. 5. Please amend either the title on the online submission form (via Edit Submission) or the title in the manuscript so that they are identical. 6. Please amend your authorship list in your manuscript file to include author "Rubina Nasir". 7. Please upload a new copy of Figures 1, 3, 9, and 10 as the detail is not clear. Please follow the link for more information: 

https://blogs.plos.org/plos/2019/06/looking-good-tips-for-creating-your-plos-figures-graphics/

https://blogs.plos.org/plos/2019/06/looking-good-tips-for-creating-your-plos-figures-graphics/  Additional Editor Comments (if provided): All the reviewers detected that the novelty of this paper is not clearly described and higlighted. Substantial additional work is needed to achieve the publication standards in terms of technical contribution and novelty. The authors are encouraged to carefully consider the reviewers' comments to improve their work.Reviewers' comments: Reviewer's Responses to Questions

**Comments to the Author**

1. Is the manuscript technically sound, and do the data support the conclusions?

Reviewer #1: Yes

Reviewer #2: Partly

2. Has the statistical analysis been performed appropriately and rigorously? 

Reviewer #1: No

Reviewer #2: Yes

3. Have the authors made all data underlying the findings in their manuscript fully available?

Reviewer #1: Yes

Reviewer #2: Yes

4. Is the manuscript presented in an intelligible fashion and written in standard English?

Reviewer #1: Yes

Reviewer #2: Yes

5. Review Comments to the Author

Reviewer #1: 1. The topic of the manuscript is interesting and significant. However, the Introduction is very poor, and the content looks too disorderly. The introduction is the comprehensive reference and deep analysis, instead of the simple list of documents. New ideas and theories (or application of those to existing problems) should be presented if they are comprehensively compared against state-of-the-art methods for many different problems and for a wide range of empirical instances in order to be convincing. Otherwise, there is little gain for the scientific community for yet another analytic method. I suggest that the Introduction should be improved and it should be also more artistic.

2. The motivation of using the proposed method is not given in this work. The novelty of the approach should be highlighted.

3. Please give the application of solar energy forecasting. And the logicality and artistic quality should be considered to present the high quality of this paper. Therefore, the presentation and the statement should be improved strongly.

Reviewer #2: Manuscript Number: PONE-D-22-24267

In this paper, long short-term memory, bidirectional LSTM, and gated recurrent unit have been used for estimating solar energy generation for every five-minute interval. My comments are:

1- The proposed method consists of some well-known methods. The proposed method novelty should be clarified.

2- Why the proposed model is useful for solar energy generation forecasting? Many accurate methods based on deep learning has been presented recently.

3- Have the weather conditions amount been forecasted for next hours? Which method? What was prediction error?

4- Which optimization method has been used for the hyper parameters optimal selection?

6. PLOS authors have the option to publish the peer review history of their article (what does this mean?). If published, this will include your full peer review and any attached files.

Reviewer #1: No

Reviewer #2: No

---

## [Author Response · Author response to Decision Letter 0]

30 Nov 2022

Comments from Editor:

a. The name of the colleague or the details of the professional service that edited your manuscript.

Response: Our Colleague and now co-author, Dr. Rizwan Khan, School of Computer Science and Mathematics, Zhejiang Normal University, Zhejiang, Jinhua, 321004, China edited the manuscript. 

b. A copy of your manuscript showing your changes by either highlighting them or using track changes (uploaded as a *supporting information* file). 

Response: Done

Response: Done

 Response: 

Code Availability Statement

The code associated with the article is available on this link https://github.com/MuhammadMuneeb007/Code---Short-term-solar-energy-forecasting-Integrated-computational-intelligence-of-LSTMs-and-GRU/blob/main/Readme.txt. The associated dataset is available on the following link: https://drive.google.com/file/d/12vtQi0zYZj6hz4jZDTXiF4Zkza42i2RQ/view?usp=sharing.

Response: Done

Please update the data availability statement as the dataset analyzed during the current study is taken from the weather station at Amherst and is available in the UMass Trace Repository, Massachusetts, USA [http://weather.cs.umass.edu/].

5. Please amend either the title on the online submission form (via Edit Submission) or the title in the manuscript so that they are identical.

Response: Done

Short-term solar energy forecasting: Integrated computational intelligence of LSTMs and GRU

6. Please amend your authorship list in your manuscript file to include author "Rubina Nasir".

Response: Done and requested another author, Dr. Rizwan Khan to be added, please.

7. Please upload a new copy of Figures 1, 3, 9, and 10 as the detail is not clear. Please follow the link for more information: 

https://blogs.plos.org/plos/2019/06/looking-good-tips-for-creating-your-plos-figures-graphics/

https://blogs.plos.org/plos/2019/06/looking-good-tips-for-creating-your-plos-figures-graphics/

Response: Done

Additional Editor Comments (if provided):

All the reviewers detected that the novelty of this paper is not clearly described and highlighted. Substantial additional work is needed to achieve the publication standards in terms of technical contribution and novelty. The authors are encouraged to carefully consider the reviewers' comments to improve their work.

Response: Novelty has been clearly stated in the revised manuscript. Substantial revision of many sections and extensive simulations have been executed to make our manuscript more effective under the guidelines and recommendations of the respected reviewers.

---

## [Decision Letter · Decision Letter 1]

19 Jan 2023

PONE-D-22-24267R1

Short-term solar energy forecasting: Integrated computational intelligence of LSTMs and GRU

PLOS ONE

Dear Dr. Zameer,

Thank you for submitting your manuscript to PLOS ONE. After careful consideration, we feel that it has merit but does not fully meet PLOS ONE’s publication criteria as it currently stands. Therefore, we invite you to submit a revised version of the manuscript that addresses the points raised during the review process.

We look forward to receiving your revised manuscript.

Kind regards,

Shuo-Yan Chou

Academic Editor

PLOS ONE

Journal Requirements:

1.  Our internal editors have looked over your manuscript and determined that it is within the scope of our Smart Energy Systems Call for Papers. The Collection will encompass the latest research in smart grid technologies, including information technologies, device integration, distribution methods, and data mining, all towards improving the efficiency of energy supply networks. Additional information can be found on our announcement page: https://collections.plos.org/call-for-papers/smart-energy-systems/. If you would like your manuscript to be considered for this collection, please let us know in your cover letter and we will ensure that your paper is treated as if you were responding to this call. If you would prefer to remove your manuscript from collection consideration, please specify this in the cover letter.

Additional Editor Comments:

After a thorough review of your revised manuscript, the editorial has determined that there are still a number of issues that need to be addressed before the paper can be accepted for publication. We appreciate the effort you have put into addressing the previous round of comments, but there are still some areas that require further attention. We would appreciate it if you could address these issues and resubmit the revised manuscript as soon as possible. Please also include a point-by-point response to the reviewers' comments in your resubmission.

Reviewers' comments:

Reviewer's Responses to Questions

**Comments to the Author**

1. If the authors have adequately addressed your comments raised in a previous round of review and you feel that this manuscript is now acceptable for publication, you may indicate that here to bypass the “Comments to the Author” section, enter your conflict of interest statement in the “Confidential to Editor” section, and submit your "Accept" recommendation.

Reviewer #1: All comments have been addressed

Reviewer #3: (No Response)

2. Is the manuscript technically sound, and do the data support the conclusions?

Reviewer #1: Yes

Reviewer #3: Partly

3. Has the statistical analysis been performed appropriately and rigorously? 

Reviewer #1: Yes

Reviewer #3: Yes

4. Have the authors made all data underlying the findings in their manuscript fully available?

Reviewer #1: Yes

Reviewer #3: Yes

5. Is the manuscript presented in an intelligible fashion and written in standard English?

Reviewer #1: Yes

Reviewer #3: Yes

6. Review Comments to the Author

Reviewer #1: 1. The abstract should not contain abbreviation. The most core framework, contribution and results should be presented in the abstract. Please gives it within one paragraph.

2. Whether the 0 value and NaN value exist in the dataset. So how do you handle it.

3. Please check the format about you title in the manuscript. ‘2.3.3 Performance Measures’ should be given in topcase writing.

4. There exists many one paragraph, which means logicality and artistic quality should be improved.

Reviewer #3: TITLE: Short-term solar energy forecasting: Integrated computational intelligence of LSTMs and GRU

Manuscript ID : PONE-D-22-24267_R1_reviewer

Type of manuscript : Original Article

Summary

In this paper, the authors carry out a comparative analysis of the performance of using Lasso regression, ridge regression, ElasticNet regression, and Support Vector Regression, as well as deep learning models for time series analysis including long short-term memory (LSTM), bidirectional LSTM (Bi-LSTM), and gated recurrent unit (GRU) along with their variants for estimating solar energy generation for every five-minute interval on Amherst weather power station.

I found this paper interesting to read and is an important paper in renewable energy modelling and forecasting. However, the manuscript requires some improvements. Some general comments are:

General Comments

In the “Abstract”, the authors should briefly present and justify the novelty of their approach in contrast to the existing body of knowledge.

Provide a list of abbreviations used in the paper.

There are some typos and grammatical errors in the manuscript.

Insert full stops at the end of the figure and table captions.

At the end of every equation which is followed by “where”, insert a comma. For example: y=x+βx, (1)

where β is a constant.

It is usually not recommended to use MAPE and R-squared as evaluation metrics when working with renewable energies such as solar power. The authors are advised to read the following paper. Hong et al.: Energy Forecasting: A Review and Outlook https://doi.org/10.1109/OAJPE.2020.3029979 See for example the section on Common Issues.

I suggest that the authors combine the forecasts from the prediction models. See for example Wang X., Hyndman R.J, Li F. and Kang Y. (2022). Forecast combinations: an over 50-year review. arXiv:2205.04216v1 and Mpfumali et al. (2019) http://dx.doi.org/10.3390/en12183569, among others. You may also want to provide the prediction intervals (PIs) for your best forecasting model. This is important as PIs capture the uncertainty surrounding the forecasts.

Usually the three shrinkage methods Lasso, ridge and elaticnet are used for variable selection after which more advanced methods are then used for forecasting. I suggest that the authors compare the predictive capabilities of SVR, LSTM, Bi-LSTM and GRU on covariates selected by the three shrinkage methods. But should clearly show their contribution to this study.

In the “Discussion” section the authors should present their findings and their main implications, also highlight the current limitations of their study and briefly mention some precise directions that they intend to follow in their future research work. Can the authors mention how much of their research is being influenced by the data they used or to which extent the methodology used within the developed research can be easily applied to other situations when the datasets differ? In this way, the authors could highlight the generalisation capability of their approach to be able to justify a wider contribution that has been brought to the current state of the art.

In the "Conclusion" section authors should avoid simply summarizing the aspects that they have already stated in the body of the manuscript. Instead, they should interpret their findings at a higher level of abstraction than in the previous sections of the manuscript. The authors should highlight whether, or to what extent they have managed to address the necessity identified within the "Introduction" section (the identified gap). The authors should avoid restating everything they did once again, but instead, they should emphasize what their findings mean to the readers, therefore making the "Conclusions" section interesting and memorable to them. The authors should not restate what they have done or what the article does, they should focus instead on what they have discovered and most importantly on what their findings mean.

7. PLOS authors have the option to publish the peer review history of their article (what does this mean?). If published, this will include your full peer review and any attached files.

Reviewer #1: No

Reviewer #3: No

---

## [Author Response · Author response to Decision Letter 1]

20 Mar 2023

Response to reviewers 

Manuscript No.: PONE-D-22-24267R1

Title: Short-term solar energy forecasting: Integrated computational intelligence of LSTMs and GRU

Authors: Aneela Zameer1#, Fatima Jaffar2, Farah Shahid3, Muhammad Muneeb4, Rizwan Khan5, Rubina Nasir 

Reviewer 1: I have incorporated all your suggestions into my revision. They were very helpful to improve the quality of our manuscript. Thank you.

Reviewer 2: I have incorporated all your suggestions into my revision. They were very helpful to improve the quality of our manuscript. Thank you.

Reviewer 3: I have incorporated all your suggestions into my revision. They were very helpful to improve the quality of our manuscript. Thank you.

---

## [Decision Letter · Decision Letter 2]

24 Apr 2023

Short-term solar energy forecasting: Integrated computational intelligence of LSTMs and GRU

PONE-D-22-24267R2

Dear Dr. Zameer,

We’re pleased to inform you that your manuscript has been judged scientifically suitable for publication and will be formally accepted for publication once it meets all outstanding technical requirements.

Kind regards,

Shuo-Yan Chou

Academic Editor

PLOS ONE

Additional Editor Comments (optional):

Reviewers' comments:

Reviewer's Responses to Questions

**Comments to the Author**

1. If the authors have adequately addressed your comments raised in a previous round of review and you feel that this manuscript is now acceptable for publication, you may indicate that here to bypass the “Comments to the Author” section, enter your conflict of interest statement in the “Confidential to Editor” section, and submit your "Accept" recommendation.

Reviewer #3: All comments have been addressed

2. Is the manuscript technically sound, and do the data support the conclusions?

Reviewer #3: Yes

3. Has the statistical analysis been performed appropriately and rigorously? 

Reviewer #3: Yes

4. Have the authors made all data underlying the findings in their manuscript fully available?

Reviewer #3: No

5. Is the manuscript presented in an intelligible fashion and written in standard English?

Reviewer #3: Yes

6. Review Comments to the Author

Reviewer #3: Dear Editor and Authors

The authors have adequately addressed my comments. I therefore recommend that the paper be accepted for publication.

7. PLOS authors have the option to publish the peer review history of their article (what does this mean?). If published, this will include your full peer review and any attached files.

Reviewer #3: No

---

## [Editor Report · Acceptance letter]

2 Jun 2023

PONE-D-22-24267R2 

Short-term solar energy forecasting: Integrated computational intelligence of LSTMs and GRU 

Dear Dr. Zameer:

I'm pleased to inform you that your manuscript has been deemed suitable for publication in PLOS ONE. Congratulations! Your manuscript is now with our production department. 

Kind regards, 

on behalf of

Professor Shuo-Yan Chou 

Academic Editor

PLOS ONE